# Towards Quantization-Aware Training for Ultra-Low-Bit Reasoning LLMs

**Yasuyuki Okoshi,**[*] **Hikari Otsuka,**[*] **Daichi Fujiki, Masato Motomura**
AI Computing Research Unit
Institute of Science Tokyo

## Abstract

Large language models (LLMs) have achieved remarkable performance across diverse reasoning tasks, yet their deployment is hindered by prohibitive computational and memory costs. Quantization-aware training (QAT) enables ultra-low-bit compression ($< 4$ bits per weight), but existing QAT methods often degrade reasoning capability, partly because complex knowledge structures are introduced during the post-training process in LLMs. In this paper, through a systematic investigation of how quantization affects different data domains, we find that its impact on pre-training and reasoning capabilities differs. Building on this insight, we propose a novel two-stage QAT pipeline specifically designed for reasoning LLMs. In the first stage, we quantize the model using mixed-domain calibration data to preserve essential capabilities across domains; in the second stage, we fine-tune the quantized model with a teacher-guided reward-rectification loss to restore reasoning capability. We first demonstrate that mixed-domain calibration outperforms single-domain calibration at maximum $2.74\%$ improvement on average over six tasks including reasoning and pre-trained tasks. Following experiments on five reasoning benchmarks show that our 2-bit-quantized Qwen3-8B outperforms post-training quantization (PTQ) baselines by $50.45\%$ on average. Moreover, compared to ultra-low-bit-specialized models such as BitNet-2B4T, our pipeline achieves approximately $2\%$ higher mathematical-reasoning accuracy with fewer than 1B tokens. Code is available: `https://github.com/yasu0001/ReasoningQAT`.

## 1 Introduction

Large language models (LLMs) have demonstrated remarkable performance across various tasks, including mathematics (Shao et al., 2024; Wang et al.; Yang et al., 2024), coding (Hui et al., 2024; Roziere et al., 2023), and knowledge-intensive question answering (Lu et al., 2022). However, their prohibitive computational and memory requirements pose significant challenges for deployment in inference. One promising direction for reducing these inference costs is weight quantization (Zhou et al., 2024; Lang et al., 2024), which employs low-bit widths for model weights. Among various quantization methods, *quantization-aware training* (QAT), which fine-tunes the model with quantized weights, is especially effective for *ultra-low-bit widths* ($< 4$ bits) (Wang et al., 2023; Ma et al., 2024; Xu et al., 2024), enabling us to deploy lightweight and fast LLMs. For example, 2-bit quantized LLMs via QAT can achieve performance comparable to their pre-quantized fp16 counterparts (Ma et al., 2024; Kaushal et al., 2024; Liu et al., 2025c).

Despite the promising performance of QAT, existing approaches suffer from severe performance degradation on reasoning benchmarks (Du et al., 2024), such as mathematics, and instruction-following tasks (Lee et al., 2025). We hypothesize that this degradation arises from the complex knowledge structures introduced during post-training. The post-training process is an extensive process that includes supervised fine-tuning (Wei et al., 2021) and preference optimization (Ouyang et al., 2022; Rafailov et al., 2023), introducing new reasoning capabilities with existing commonsense knowledge acquired during pre-training. While it creates heterogeneous knowledge structures, it remains unclear how quantization affects the model's performance on reasoning capabilities and pre-trained commonsense knowledge.

---

[*]Equal contribution

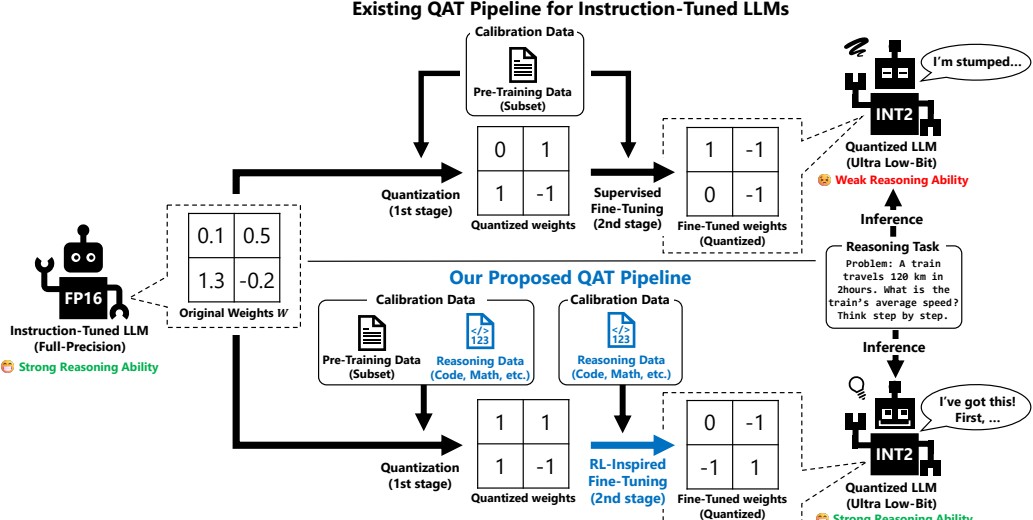

Figure 1: Comparison of the existing QAT pipeline with the proposed pipeline.

To address this gap, we conduct a systematic investigation of how quantization impacts different knowledge domains in post-training LLMs. Our analysis reveals that quantization creates inherent trade-offs between commonsense knowledge preservation and reasoning capability retention, where different domains exhibit varying sensitivity to quantization. Specifically, while performance on commonsense knowledge benchmarks remains relatively stable even with quantization using out-of-domain data, reasoning capabilities exhibit significant sensitivity to quantization data, suggesting that different domains have distinct requirements for effective quantization.

Based on this analysis, we introduce a quantization framework specifically designed for post-trained LLMs that address diverse knowledge domains through a novel two-stage pipeline. Following our observation, our quantization framework is designed to dedicate computational resources to maintain reasoning capability, with minimal efforts to preserve general knowledge. Specifically, the first stage carries out block-wise quantization with mixed-domain calibration. This mixed-domain calibration preserves essential reasoning capabilities that are difficult to restore, while also maintaining commonsense knowledge. Subsequently, we perform end-to-end fine-tuning with reinforcement learning inspired objectives to enhance reasoning capability. This unified framework enables extremely low-bit quantization of post-trained LLMs with minimal reasoning performance degradation.

Extensive experiments on five reasoning benchmarks demonstrate the effectiveness of our approach. Our method achieves significant improvements over existing post-training quantization methods for reasoning LLMs. Specifically, our 2-bit quantized Qwen3-8B outperforms other quantization methods by $50.45\%$ on average. Notably, even when compared to specialized ternary LLMs like BitNet-2B4T, our 2-bit model with 1.7B parameters demonstrates superior mathematical reasoning performance with substantially reduced training costs—achieving $2.5\%$ improvement using fewer than 1B training tokens.

Our contributions can be summarized as follows:

- We empirically demonstrate how quantization differently affects commonsense knowledge acquired during pre-training and reasoning capabilities developed in post-training. Our results highlight the importance of designing mixed calibration data to effectively preserve both of them.

- We propose two-stage quantization pipeline for post-trained LLMs that combines mixed-domain calibration and RL-inspired fine-tuning to preserve reasoning capabilities while achieving extremely low-bit quantization.

- We demonstrate that our approach achieves state-of-the-art (SOTA) performance on multiple reasoning benchmarks with both 2-bit and 3-bit quantization.

## 2 PRELIMINARIES

This section outlines the weight quantization and recent quantization-aware training (QAT) pipeline.

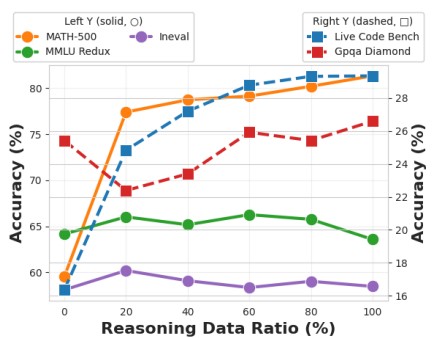 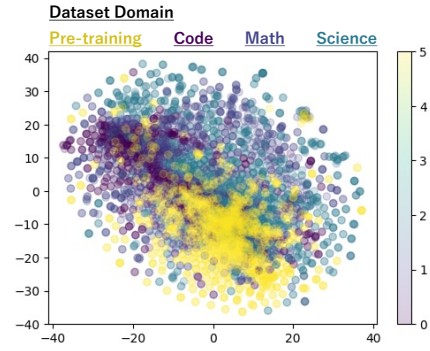

(a) Performance for general (green ● and purple ●) and mathematical (orange ●, blue ■, and red ■) tasks on the reasoning data ratio.

(b) t-SNE visualization of the activations in the 14th Transformer block, where colors denote dataset domains.

Figure 2: Impact of data-domain composition on Qwen3-1.7B. In the ultra-low-bit quantized model, replacing a part of the calibration data drawn from the pre-training data (FineWeb-Edu) with reasoning data (OpenThoughts3-1.2M) leaves general-task accuracy unchanged while improving reasoning accuracy (left). In the full-precision model, t-SNE of the model's activations shows tight clusters for pre-training inputs but wide dispersion for reasoning inputs (right).

**Weight Quantization:** Weight quantization maps the model weights to low-bit width counterparts. Given a full-precision weight $w \in \mathbb{R}$, we obtain its dequantized approximation $w'$ via

$$q := \mathrm{clamp}\left(\lfloor w/s \rceil + z, 0, 2^N - 1\right), \qquad w' := s\left(q - z\right),$$

where $s > 0$ is the scale factor, $z \in \mathbb{R}$ is the zero point, and $N$ is the target bit width. $\lfloor \cdot \rceil$ represents the nearest integer function (i.e., the round function), and $\mathrm{clamp}(x, a, b)$ clamps input $x$ to the interval $[a, b]$. Since the scale $s$ and zero point $z$ are shared across groups of weights (e.g., an entire matrix, a channel, or a block), each weight is represented only by an $N$-bit code $q$, with $s$ and $z$ stored once per group, achieving a low-bit width per-weight representation.

Weight quantization approaches are mainly categorized into two strategies: 1) post-training quantization (PTQ), which converts pre-trained model weights to low-bit widths without retraining; and 2) quantization aware training (QAT), which quantizes and fine-tunes the weights simultaneously. PTQ can quickly quantize weights with small or even without calibration data, while it struggles with ultra-low-bit quantization. In contrast, QAT can flexibly fine-tune the full-precision weights $w$, scaling factor $s$, and zero point $z$, achieving performance comparable to the full-precision model in ultra-low-bit scenarios.

**Quantization-Aware Training Pipeline:** As illustrated in the top panel of Figure 1, existing QAT pipelines generally comprise two stages: 1) an initial quantization stage; and 2) a fine-tuning stage. The first stage initializes the quantized weights that serve as the starting point for the subsequent stage. Some methods omit this step, whereas the latest state-of-the-art (SOTA) QAT approaches (Chen et al., 2024; Du et al., 2024) have demonstrated that quantizing weights using a subset of the pre-training dataset as calibration data enables stable fine-tuning during the subsequent stage. In the second stage, the weights quantized in the first stage are fine-tuned by minimizing a training objective. All parameters (i.e., weights, scale, zero point) or some of them are fine-tuned, and this paper fine-tunes only the scale, following one of the SOTA QAT approaches, EfficientQAT. Also, the training objective is typically either a self-supervised pre-training loss (Liu et al., 2025c; Chen et al., 2024) or a knowledge-distillation loss (Du et al., 2024; Lee et al., 2025).

## 3 REASONING-ORIENTED TWO-STAGE QUANTIZATION AWARE TRAINING

This section proposes a novel QAT pipeline that enables the preservation of reasoning capabilities after ultra-low-bit quantization. We first analyze the impact of quantization on various knowledge domains, and based on these findings, we introduce the reasoning-oriented QAT pipeline.

## 3.1 Quantization Impacts Across Knowledge Domains

This section analyzes how domain selection for calibration data affects the overall model performance. Existing quantization approaches mainly perform quantization with either pre-training data (Liu et al., 2025c; Chen et al., 2024), or domain-specific data, such as mathematics (Liu et al., 2025a). While previous work has selected calibration data tailored to specific target tasks, the cross-task implications of such task-specific calibration choices remain largely unexplored.

To investigate the effect of domain selection for calibration data, we analyze the impact of selecting different calibration datasets on performance across multiple tasks and knowledge domains. As shown in the results of Qwen3-1.7B quantized to 3-bits by EfficientQAT (Chen et al., 2024) (Figure 2a), the tasks can be broadly grouped into two trends: 1) tasks for which performance improves as the amount of reasoning data increases; and 2) tasks for which performance remains almost constant regardless of the amount of reasoning data. Notably, all tasks in the first category are represented in the reasoning dataset, i.e., code (blue), mathematical tasks (orange), scientific questions and answers (red). These results indicate that reasoning data tends to suffer from domain shift, while tasks related to commonsense knowledge are less sensitive to calibration datasets. On the other hand, common tasks also demonstrate performance degradation when calibrated with pure reasoning data, suggesting the importance of dataset diversity even for tasks that appear less sensitive to calibration choices.

These distinct trends happen as the intermediate distributions between pre-trained data and reasoning data differ, as shown in Figure 2b. The distributional mismatch leads to suboptimal quantization performance when calibrating on single-domain data, resulting in higher quantization errors for tasks that require domain-specific representations.

## 3.2 Proposed Method

We now introduce the novel QAT pipeline for ultra-low-bit reasoning LLMs.

**Knowledge Domain Selection in Calibration data:**   We first focus on the mixing ratio of the knowledge domain in calibration data. The results in Section 3.1 illustrate the importance of selecting appropriate calibration data when a QAT pipeline is applied to post-trained LLMs. In particular, it is important to mix pre-training data and reasoning data in an appropriate ratio. Building on these findings, we propose using novel calibration data in the first stage of the QAT pipeline. This data is composed of $80\%$ reasoning-focused data and $20\%$ pre-training data, designed to bias the calibration process toward reasoning while retaining coverage of pre-training distributions.

**Supervised Fine-Tuning With Reward Rectification Loss:**   We secondly aim at the fine-tuning stage. Quantization of the first stage using proposed calibration data mixed with pre-training data preserves the fundamental capabilities of the LLM, enabling us to focus on enhancing reasoning capabilities during the fine-tuning stage. A straightforward approach to enhance reasoning ability is to perform supervised fine-tuning using reasoning data, but such training does not effectively generalize into reasoning data (Chu et al., 2025). Employing reinforcement learning could improve generalization on reasoning data, but online text generation incurs auto-regressive text generations, resulting in huge training overhead. To balance training efficiency and generalization on unseen data, we employ reweighted rectification (Wu et al., 2025) for supervised fine-tuning to make the objective function reinforcement-like.

Reward rectification is scaling factors for the loss function in supervised fine-tuning. Given the datasets $\mathcal{D} = \{x, y^*\}$ and the supervised fine-tuning loss $\mathcal{L}_{\mathrm{SFT}}(\theta)$, reward rectification loss $\mathcal{L}(\theta)$ dynamically reweights the supervised loss as follows:

$$\mathcal{L}(\theta) = \mathcal{L}_{\mathrm{SFT}}(\theta) \cdot \mathrm{sg}(1/w),$$

where $w$ is the dynamic reweighting factor and $\mathrm{sg}(\cdot)$ denotes the stop-gradient operator.

This formulation can be viewed as bridging supervised fine-tuning and reinforcement learning. In particular, choosing $w = 1/\pi_\theta(y \mid x)$ yields a gradient equivalent to an on-policy policy-gradient update with the reward function:

$$r(x, y) = \mathbf{1}[y = y^*],$$

where $\pi_\theta(y \mid x)$ is the model's conditional probability of generating an output $y$ given an input $x$ under parameters $\theta$. This dynamic re-weighting can avoid over-concentration on low-probability reference tokens, improving generalization despite not using additional sampling or reward functions.

While the original reward rectification uses the student model's own probability $\pi_\theta(y \mid x)$ for reweighting, in the QAT, the quantized model's distribution becomes less reliable due to precision loss. Using the quantized model's own probabilities for reweighting could amplify these errors.

Therefore, we leverage the teacher model's probability $\pi_t(y^*|x)$ as a more reliable reference for the reweighting factor. This teacher-guided approach ensures that the reweighting process is based on the target distribution we aim to recover, rather than the potentially corrupted distribution of the quantized model.

Thus, we introduce teacher-guided reward rectification loss $\mathcal{L}(\theta)$, where the teacher model $\pi_t$ controls the scale of supervised loss function. Given the teacher probability with labeled data $\pi_t(y^* \mid x)$, teacher guided reward rectification loss can be represented as:

$$\mathcal{L}_t(\theta) = \mathcal{L}_{\mathrm{SFT}}(\theta) \cdot \mathrm{sg}(\pi_t(y^*|x)).$$

Intuitively, this formulation represents that the supervised loss values are amplified when the probability of the quantized model for the label is smaller than that of the teacher probability. When the distribution of the quantized model becomes close to the original distribution, this scaling factor acts as the original reward rectification.

To align the overall probabilistic distribution of the quantized model with original LLMs, we further introduce an additional KL divergence loss. Finally, our training loss function can be represented as:

$$\mathcal{L}(\theta) = \alpha \mathcal{L}_t(\theta) + \beta D_{\mathrm{KL}}\big(\pi_{\mathrm{T}}(\cdot|x)||\pi_{\mathrm{S}}(\cdot|x)\big), \tag{1}$$

where $D_{\mathrm{KL}}\big(\pi_{\mathrm{T}}(\cdot|x)||\pi_{\mathrm{S}}(\cdot|x)\big) = \sum_y \pi_{\mathrm{T}}(y|x) \log \frac{\pi_{\mathrm{T}}(y|x)}{\pi_{\mathrm{S}}(y|x)}$ is the KL divergence between the fp16 model and the quantized model, and $\alpha, \beta$ is hyperparameters that control the effects of teacher-guided reward rectification loss and kl divergence loss.

# 4 EXPERIMENTS

## 4.1 EXPERIMENTAL SETTINGS

**Training:** We conduct experiments on Qwen3 instruction-tuned models (Yang et al., 2025). For block-wise calibration, we use a total $4,096$ samples with a context length of $2,048$. Calibration datasets consist of $80\%$ of sequences sampled from OpenThoughts-1.2M (Guha et al., 2025) and the remaining $20\%$ sampled from FineWeb-Edu (Lozhkov et al., 2024). We use different learning rates for quantization parameters (1e-4) and weight parameters (1e-5). For 2-bit quantization, we use a larger learning rate of 2e-5 for weights.

During supervised fine-tuning, models are with $32,768$ samples from OpenThoughts-1.2M. We optimize all trainable parameters with the same learning rate. The learning rate for 3-bit quantization is 1e-6, while we use a larger learning rate for 2-bit quantization, 5e-6 for the 1.7B parameter, and 1e-4 for other parameters. We use the AdamW optimizers (Loshchilov & Hutter, 2019) with the cosine annealing learning rate decay (Loshchilov & Hutter, 2017). Models are fine-tuned with a batch size of $64$ and one epoch for 3-bit models, except for Qwen 1.7B at 2-bit which uses 3 epochs. We filter out the top-20 probabilities for the KL loss. We set $\alpha = 0.2$ and $\beta = 1.0$ in Equation (1) unless explicitly stated otherwise.

**Evaluation:** We evaluate the zero-shot accuracy on five benchmarks including We evaluate the zero-shot accuracy on five benchmarks, including MATH-500 (Lightman et al., 2023), Live Code Bench (White et al., 2024), MMLU-Redux (Gema et al., 2024), GPQA- Diamond (Rein et al., 2024), and IFEval (Zhou et al., 2023), using the evalscope (Team, 2024). These tasks are evaluated in open-ended text generation. We use token-level sampling, whose tokens are sampled from the top 20 highest tokens with a temperature of 0.6. We basically use a maximum sequence length of

Table 1: Accuracy comparison for different calibration data on 6 benchmarks. Higher values are better. We define the group size as 128. Mixed data contains 80% of reasoning data and 20% of pre-training data.

| Model (Qwen3) | Bit Width | Dataset Type | Reasoning Tasks | | | Pre-trained Tasks | | | |
|---|---|---|---|---|---|---|---|---|---|
| | | | MATH-500 | Live Code Bench | GPQA-Diamond | MMLU-Redux | CSR | IFEVAL | Avg. |
| 1.7B | w3 | Pre-training | 59.53 | 16.36 | 25.42 | 64.16 | **57.96** | 58.60 | 47.01 |
| | | Reasoning | **81.33** | **29.35** | **26.60** | 63.58 | 55.82 | 59.52 | 52.70 |
| | | Mixed | 80.20 | 29.32 | 25.42 | **65.76** | 56.71 | **59.70** | **52.85** |
| | w2 | Pre-training | 0.13 | 0.00 | 0.00 | 0.00 | 50.75 | 13.86 | 10.79 |
| | | Reasoning | **20.60** | 0.09 | 7.58 | 19.93 | 44.77 | **25.51** | 19.75 |
| | | Mixed | 18.68 | **0.47** | 7.58 | 28.92 | 49.11 | 24.58 | **21.56** |
| 4B | w3 | Pre-training | 81.80 | 35.42 | 41.92 | 77.98 | 63.94 | 71.16 | 62.04 |
| | | Reasoning | **90.90** | **46.89** | 42.76 | 78.55 | 63.71 | 71.72 | 65.76 |
| | | Mixed | 90.70 | 46.85 | **45.45** | **78.91** | **63.97** | 74.86 | **66.79** |
| | w2 | Pre-training | 2.73 | 0.00 | 6.23 | 26.89 | **59.34** | 17.74 | 18.82 |
| | | Reasoning | **33.80** | **5.78** | 11.62 | 46.26 | 53.41 | **32.90** | **30.63** |
| | | Mixed | 22.60 | 5.12 | **14.14** | **51.20** | 55.89 | 31.05 | 30.00 |
| 8B | w3 | Pre-training | 87.00 | 37.25 | 41.66 | 82.60 | **69.19** | 76.34 | 65.67 |
| | | Reasoning | 91.80 | **53.84** | **51.52** | 81.75 | 68.14 | 75.79 | 70.47 |
| | | Mixed | **92.40** | 51.75 | 48.99 | **83.33** | 69.16 | 81.15 | **71.13** |
| | w2 | Pre-training | 5.33 | 0.28 | 4.55 | 41.72 | **63.35** | 19.41 | 22.44 |
| | | Reasoning | **42.27** | **8.15** | 10.44 | 51.40 | 55.05 | 35.86 | 33.86 |
| | | Mixed | 40.00 | 7.30 | **11.11** | **57.11** | 60.81 | **43.25** | **36.60** |

32K for all benchmarks. However, we reduce the maximum sequence length to 8K on the lower-performance models to avoid excessive text generation due to the absence of a stop token. All evaluations are conducted three times, and we report the average accuracy.

**Quantization baselines to post-trained LLMs:** We compare our method with two PTQ quantization baselines, GPTQ (Frantar et al., 2022) and AWQ (Lin et al., 2024), both of which are evaluated on reasoning benchmarks (Liu et al., 2025a). To quantize these two baselines, we follow a similar strategy as conducted by Liu et al. (2025a). Specifically, we perform quantization using 128 samples from NuminaMath (LI et al., 2024). We reproduce these quantized models locally, except for 3 and 4 bits AWQ quantization for Qwen-8B, as reproduced performance is much inferior to the performance claimed in the paper (Liu et al., 2025a).

### 4.2 DATASET EFFECTS ON OVERALL PERFORMANCE

This section evaluates our proposed calibration datasets against single-domain calibrations using either pre-training data or reasoning data. We evaluate five benchmarks described in Section 4.1 with the additional commonsense reasoning tasks (CSR) to evaluate general knowledge of quantized models. The CSR includes five subset tasks: ARC-e, ARC-c (Clark et al., 2018), PIQA (Bisk et al., 2020), HellaSwag (Zellers et al., 2019), and WinoGrande Sakaguchi et al. (2021). Table 1 demonstrates that mixed domain calibration outperforms single-domain calibration across various parameters in both 2-bit and 3-bit quantization settings. Compared with pre-training datasets, its performance improvements on reasoning benchmarks, including mathematical and coding tasks, are particularly notable. In addition, mixed data achieves comparable performance to pre-trained data and other benchmarks in reasoning tasks. When compared with reasoning-only datasets, the performance of mixed datasets on reasoning benchmarks is close, while performance in pre-trained domain benchmarks tends to be superior. These results suggest that including a small portion of pre-training data can be effective in maintaining commonsense knowledge. resulting in better generalization results across a wide range of tasks.

### 4.3 COMPARISON WITH PRIOR QUANTIZATION APPROACHES FOR REASONING

This section compares our proposal with other quantization methods, including GPTQ and AWQ. Table 2 shows that our proposal significantly outperforms existing quantization approaches in both

Table 2: Comparison of quantization methods and bit-widths on Qwen3 models. Values are % (higher is better). We define the group size as 128.

| Settings | | | Benchmarks (%) | | | | | Avg. |
|---|---|---|---|---|---|---|---|---|
| Model | Method | Bit Width(W/A) | MATH500 | LiveCodeBench | MMLU-Redux | GPQA-Diamond | IFEval | Avg. |
| **Qwen3-1.7B** | | | | | | | | |
| | FP (Baseline) | bfloat16 | 89.0 | 53.6 | 74.7 | 38.4 | 70.4 | 65.2 |
| | GPTQ | 4/16 | 86.3 | 36.9 | 70.2 | 34.3 | 66.2 | 58.8 |
| | AWQ | 4/16 | 87.4 | 44.4 | 71.6 | 35.9 | 65.3 | 60.9 |
| | GPTQ | 3/16 | 58.2 | 0.0 | 42.4 | 9.3 | 31.8 | 28.3 |
| | AWQ | 3/16 | 58.0 | 5.8 | 53.5 | 17.5 | 47.7 | 36.5 |
| | **Proposal** | 3/16 | **82.7** | **33.0** | **67.7** | **31.7** | **61.0** | **55.2** |
| | GPTQ | 2/16 | 2.1 | 0.0 | 5.8 | 4.7 | 8.5 | 4.2 |
| | AWQ | 2/16 | 0.0 | 0.0 | 27.3 | 8.1 | 12.3 | 9.5 |
| | **Proposal** | 2/16 | **48.6** | **6.5** | **40.1** | **14.5** | **32.2** | **28.4** |
| **Qwen3-4B** | | | | | | | | |
| | FP (Baseline) | bfloat16 | 93.6 | 71.2 | 84.3 | 51.5 | 83.6 | 76.8 |
| | GPTQ | 4/16 | 93.4 | 66.2 | 82.1 | 50.7 | 81.2 | 74.7 |
| | AWQ | 4/16 | 93.0 | 65.7 | 83.0 | 50.2 | 81.0 | 75.6 |
| | GPTQ | 3/16 | 85.0 | 21.2 | 65.6 | 24.9 | 54.7 | 50.3 |
| | AWQ | 3/16 | 88.3 | 37.2 | 74.2 | 33.7 | 71.9 | 61.1 |
| | **Proposal** | 3/16 | **89.5** | **50.2** | **79.8** | **46.8** | **75.3** | **68.3** |
| | GPTQ | 2/16 | 3.7 | 0.0 | 7.5 | 8.9 | 8.5 | 5.7 |
| | AWQ | 2/16 | 0.0 | 0.0 | 0.0 | 0.0 | 11.8 | 2.4 |
| | **Proposal** | 2/16 | **77.1** | **19.5** | **61.7** | **26.9** | **50.1** | **47.1** |
| **Qwen3-8B** | | | | | | | | |
| | FP (Baseline) | bfloat16 | 94.0 | 73.0 | 87.3 | 61.6 | 86.5 | 80.5 |
| | GPTQ | 4/16 | 94.6 | 69.6 | 86.8 | 58.1 | 86.9 | 79.2 |
| | AWQ | 4/16 | 97.0 | 54.7 | N/A | 59.6 | N/A | N/A |
| | GPTQ | 3/16 | 92.1 | 39.4 | 79.0 | 46.8 | 76.9 | 66.8 |
| | AWQ | 3/16 | **92.9** | 35.3 | N/A | 46.8 | N/A | N/A |
| | **Proposal** | 3/16 | 91.5 | **60.0** | **84.5** | **47.5** | **78.8** | **72.5** |
| | GPTQ | 2/16 | 2.8 | 0.0 | 6.4 | 5.2 | 8.7 | 4.6 |
| | AWQ | 2/16 | 0.0 | 0.0 | 5.9 | 3.9 | 10.2 | 4.0 |
| | **Proposal** | 2/16 | **80.4** | **28.5** | **72.6** | **34.5** | **59.3** | **55.1** |

2-bit and 3-bit quantization settings. The performance improvements are particularly notable in 2-bit quantization settings. While the performance of GPTQ or AWQ quantized models is extremely low, our quantized models not only achieve solid performance but also exhibit steadily increasing accuracy as the number of parameters grows. This trend suggests that our quantization method scales effectively with model size even for extremely low-bit quantization. For 3-bit quantization, our approach dramatically improves performance, particularly on smaller models. For example, our 3-bit quantization of Qwen3-1.7B achieves an average accuracy of $55.20\%$ across five tasks, which is $18.71\%$ higher than existing PTQ methods. These results highlight that our approach is especially effective when model capacity is constrained, such as in cases of ultra-low bit widths or limited parameter counts.

## 4.4 COMPARISON WITH QAT APPROACHES

This section compares our approach with two SOTA quantization-aware training (QAT) approaches: BitDistiller (Du et al., 2024) and EfficientQAT (Chen et al., 2024). To ensure a fair comparison in reasoning benchmarks, we reproduce these QAT approaches in our proposed framework. Specifically, for both methods, we perform calibration using mixed dataset, and fine-tuning with the same number of training tokens sampled from OpenThoughts-1.2M Datasets. As shown in Table 3, our approach achieves better performance in both 2-bit and 3-bit than two QAT baselines. These results demonstrate that our approach offers a more effective strategy for low-bit quantization compared to conventional QAT methods.

## 4.5 ABLATION STUDY

There are very few quantization-aware (QAT) training approaches that can be directly compared to ours, as most existing methods target different evaluation settings. Instead, this section presents ablation studies that compare our approach with key components derived from existing QAT methods.

Table 3: Comparison of SOTA quantization approaches with various bit-width on Qwen3-1.7B models. Values are % (higher is better). We define the group size as 128.

| Settings | | | Benchmarks (%) | | | | | Avg. |
|---|---|---|---|---|---|---|---|---|
| Model | Method | Bit Width (W/A) | MATH500 | LiveCodeBench | MMLU-Redux | GPQA-Diamond | IFEval | Avg. |
| **Qwen3-1.7B** | | | | | | | | |
| | FP (Baseline) | bfloat16 | 89.0 | 53.6 | 74.7 | 38.4 | 70.4 | 65.2 |
| | EfficientQAT | 3/16 | 80.8 | 29.8 | 66.2 | 30.3 | 60.4 | 53.5 |
| | BitDistiller | 3/16 | 59.4 | 10.2 | 56.6 | 17.7 | 52.3 | 39.2 |
| | **Proposal** | 3/16 | **82.7** | **33.0** | **67.7** | 31.7 | **61.0** | **55.2** |
| | EfficientQAT | 2/16 | 24.8 | 0.6 | 29.1 | 12.1 | 25.1 | 18.3 |
| | BitDistiller | 2/16 | 12.2 | 1.0 | 22.4 | **21.7** | 19.0 | 15.2 |
| | **Proposal** | 2/16 | **48.6** | **6.5** | **40.1** | 14.5 | **32.2** | **28.4** |

Table 4: Ablation on block-wise calibration and loss choice: "S" denotes conventional supervised fine-tuning; "R" denotes our proposed loss; and "C" indicates the use of block-wise calibration.

| Training | S | R | C | C+S | C+R |
|---|---|---|---|---|---|
| **MATH-500** | 1.4 | 1.60 | 28.57 | 22.70 | **38.13** |
| **Live Code Bench** | 0.0 | 0.0 | 0.47 | 0.00 | **5.75** |
| **MMLU Redux** | 3.54 | 3.53 | 28.57 | 35.78 | **36.64** |
| **GPQA Diamond** | 6.06 | 6.06 | 7.58 | 14.31 | **14.14** |
| **IFEval** | 10.72 | 12.20 | 24.58 | 23.66 | **31.61** |

Table 5: Benchmark comparison of our proposal with INT2 Qwen3 family quantized with our proposal and BitNet b1.58 2B. We represent Instruct Strict as IS.

| Benchmark (Metric) | Qwen3 8B-int2 (Ours) | Qwen3 4B-int2 (Ours) | Qwen3 1.7B-int2 (Ours) | BitNet b1.58 2B |
|---|---|---|---|---|
| **Training Tokens** | 328M | 328M | 968M | 4T |
| **Activation** | bf16 | bf16 | bf16 | int8 |
| **MATH-500** | 80.13 | 77.13 | 48.60 | 43.40 |
| **GSM8K** | 88.93 | 81.71 | 57.47 | 58.38 |
| **IFEval (IS)** | 59.33 | 50.09 | 45.29 | 53.48 |
| **Average** | 76.13 | 69.91 | 50.75 | 51.75 |

**Effectiveness of Block-wise Calibration:** The main differences between our approach and existing methods are the use of block-wise calibration before fine-tuning and the loss function. To analyze the effects of these two components, we either replace the loss function with conventional cross-entropy loss, which is basically used in QAT (Liu et al., 2025c). In these experiments, we fine-tune the 2-bit quantized Qwen-3 1.7B model for one epoch using 32K sequences with a learning rate of 5e-6. Table 4 summarizes the results. Here, "S" denotes the conventional supervised fine-tuning with cross entropy loss function, "R" denotes the teacher-guided reward rectification loss in Section 3.2, and "C" means the existence of a calibration stage. Therefore, "C+S" denotes supervised fine-tuning after calibration. As shown in Table 4, both calibration data and proposed loss function significantly enhance model performance. We also find that modifying the supervised loss led to substantial improvements on reasoning benchmarks. In particular, on MATH-500, the accuracy increased by $15.43\%$ when moving from cross entropy loss to our proposed loss, and on Live Code Bench, it increased by $5.75\%$. More importantly, the performance on reasoning benchmarks degrades after conducting supervised fine-tuning with cross-entropy loss. These results indicate that conventional QAT approaches, which rely primarily on cross-entropy loss for supervised fine-tuning, are insufficient for post-trained LLMs.

**Effectiveness of Loss Weighting:** This section studies the contribution of two loss terms, the teacher-guided reward-rectification loss and the KL-divergence loss, to overall performance. We fine-tune a 3-bit quantized Qwen3-1.7B for a single epoch with different $\alpha$ and $\beta$. We evaluate three different weighting schemes: $(\alpha, \beta) = (1, 0)$, which applies only the reward rectification loss; $(\alpha, \beta) = (0, 1)$, which applies only the KL divergence loss; and $(\alpha, \beta) = (0.2, 1)$ (i.e., our proposed configuration), which combines both losses with the specified weights. Table 6 demonstrates that combining the two losses improves overall model performance. These results demonstrate that our approach exhibits stronger gains in more aggressive quantization settings, particularly at 2-bit.

## 4.6 COMPARISON WITH BITNET1.58 2B4T

This section compares our quantized models with BitNet1.58 2B4T, a native ternary LLM trained from scratch. To align the bit-width, our QAT pipeline quantizes Qwen3 models into 2-bit.

Table 6: Effect of the loss function on reasoning performance. Comparison of 3-bit and 2-bit quantization on Qwen3 1.7B.

| Bit Width | $(\alpha, \beta)$ | MATH-500 | Live Code Bench | MMLU Redux | GPQA Diamond | IFEval |
|---|---|---|---|---|---|---|
| | $(1.0, 0.0)$ | 78.2 | 28.5 | 66.1 | 21.7 | 60.6 |
| 3 | $(0.0, 1.0)$ | **82.8** | 32.4 | 67.0 | 30.3 | **63.0** |
| | $(0.2, 1.0)$ | 82.7 | **33.0** | **67.7** | **31.7** | 62.1 |
| | $(1.0, 0.0)$ | 22.8 | 1.5 | 32.7 | **15.7** | 25.0 |
| 2 | $(0.0, 1.0)$ | 2.2 | 0.0 | 6.6 | 2.0 | 8.0 |
| | $(0.2, 1.0)$ | **48.6** | **6.5** | **40.1** | 14.5 | **32.2** |

Table 5 describes the accuracies on two mathematical benchmarks including MATH-500 and GSM8K, and IFEval. We referred to the results of BitNet1.58 2B4T from (Ma et al., 2025). As shown in Table 5, our INT2 quantized model achieves superior mathematical performance with lower parameter requirements and significantly fewer tokens required for the quantization process. These results demonstrate that by designing an appropriate QAT pipeline, it is possible to leverage pre-trained features, leading to promising reasoning performance train high-accuracy 2-bit models at a fraction of the training costs.

In addition, our approach demonstrates superior scalability compared to BitNet 1.58 2B4T. Because we fine-tune pre-trained LLMs using only a limited number of sequences, we can easily produce models with different parameter counts. This enables a flexible trade-off between performance and resource usage, as illustrated in Table 5, which demonstrates results of several parameter variations of our quantized models.

## 5 RELATED WORKS

In this section, we briefly summarize the quantization approaches. Quantization approaches can be categorized into post-training quantization (PTQ) and quantization-aware training (QAT) depending on whether fine-tuning is performed or not. This section deals with weight-only quantization of large language models (LLMs) addressed in this work.

**Post-training quantization** (PTQ) converts full-precision weights into lower-bit counterparts without relying on fine-tuning. To obtain better quantization parameters, recent methods optimize the reconstruction problem either at the linear projection level (Frantar et al., 2022; Lin et al., 2024) or at the transformer block level (Lee et al., 2023; Shao et al., 2023). While PTQ has achieved strong initial success in LLMs, initial approaches still face limitations in achieving extremely low-bit quantization without losing their performance. To overcome these challenges, research has shifted toward more aggressive quantization, such as 3-bit or 2-bit. Some approaches target such low-bit quantization with integer representation (Shao et al., 2023; Zhao et al., 2024; Chee et al., 2023), demonstrating noticeable performance at these bit-widths. To further improve the trade-offs between accuracy and model size, recent approaches introduce vector quantization (Egiazarian et al., 2024; Tseng et al., 2024; Malinovskii et al., 2024). Despite their promising performance, vector quantization introduces substantial overhead in inference (Gong et al., 2024).

**Quantization-aware training** (QAT), in contrast, can enhance quantized model performance by incorporating fine-tuning. With the additional computational cost for fine-tuning, QAT enables the use of hardware-friendly numerical representations, such as integers, for low-bit quantization, resulting in minimal overhead at inference time. There are several choices for optimization targets for fine-tuning. LLM-QAT (Liu et al., 2023) and BitDistiller (Du et al., 2024) explore knowledge distillation within QAT literature. BitNet b1.58 (Ma et al., 2024), Spectra (Kaushal et al., 2024), and ParetoQ (Liu et al., 2025c) employ fine-tuning in a self-supervised manner using pre-training data. By spending billions of tokens for fine-tuning, these approaches realize promising performance with ternary or 2-bit. Given the substantial training costs of these approaches, recent work has focused on improving the training efficiency of QAT approaches. EfficientQAT (Chen et al., 2024) introduces two two-stage pipeline that perform end-to-end backpropagation following block-wise calibration. UPQ (Lee et al., 2025) modifies the two-stage QAT pipeline to use knowledge distillation and progressive quantization, demonstrating the promising performance on instruction-tuned LLMs.

However, most existing quantization approaches have primarily focused on pre-training LLMs, with limited exploration of their effectiveness on complex reasoning capabilities that are crucial for modern LLM applications. In this paper, we investigate how quantization affects reasoning performance and propose methods to preserve reasoning capabilities in quantized LLMs.

**Quantization and Reasoning.** Several comprehensive analyses have explored the effects of quantization on reasoning capability. Li et al. (2025) and Liu et al. (2025b) demonstrate that ultra-low-bit quantization leads to severe performance drops on reasoning benchmarks such as mathematical tasks. Liu et al. (2025b) demonstrates that less-than-4-bit quantization leads to severe performance drops on reasoning benchmarks such as mathematical tasks. Mekala et al. (2025) systematically analyzes the effects of quantization on long-context reasoning tasks, demonstrating that even 4-bit models incur substantial losses. Although these analyses reveal critical challenges for existing quantization approaches, few studies have explored effective strategies to maintain reasoning performance under such aggressive settings. However, one notable example is BitNet 2B4T (Ma et al., 2025), which demonstrates strong performance on mathematical benchmarks with ternary LLMs by performing quantization-aware training over four trillion tokens. In this paper, we explore a more efficient approach for reasoning-oriented LLMs. By combining block-wise quantization with RL-inspired fine-tuning using limited tokens, we obtain highly accurate 2- and 3-bit LLMs with significantly fewer fine-tuning sequences.

## 6 CONCLUSION

This paper addresses the critical challenge of maintaining reasoning capabilities in ultra-low-bit quantized large language models (LLMs). Through systematic analysis, we demonstrate that quantization affects different knowledge domains unevenly—while pre-training knowledge remains robust, reasoning capabilities show severe degradation. Building on this insight, we develop a novel two-stage quantization-aware training pipeline specifically designed for post-trained reasoning LLMs. Our approach combines mixed-domain calibration with teacher-guided reward rectification loss to preserve and restore reasoning abilities under aggressive quantization. Experiments across five reasoning benchmarks validate our method, with 2-bit quantized Qwen3-8B achieving $50.45\%$ average improvement over existing approaches. Notably, our method outperforms BitNet 2B4T on mathematical reasoning while requiring dramatically fewer training resources. We establish the first quantization framework specifically targeting reasoning-oriented LLMs, providing practical solutions for efficient model compression without sacrificing cognitive capabilities. This work provides a foundation for future developments in efficient, high-performance quantized reasoning models, enabling broader deployment of sophisticated AI systems.

## ACKNOWLEDGMENT

This work was carried out using the TSUBAME4.0 supercomputer at Institute of Science Tokyo. This work was supported in part by JSPS KAKENHI Grant Numbers JP23H05489, JP25K03092, and JP23KJ0955, and by JST-ALCA-Next Japan Grant # JPMJAN24F3.

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

Table 7: Comparison of training cost and reasoning performance of Qwen3-1.7B under 3-bit and 2-bit quantization. GPU hours are measured using the H100 GPU.

| 2nd Step | Bit Width | GPU Hours | MATH-500 | LiveCodeBench | GPQA-Diamond | MMLU-Redux | IFEVAL | Avg. |
|----------|-----------|-----------|----------|---------------|--------------|------------|--------|------|
| GRPO (RL) | 3 | $\approx 220$ | 83 | 31.5 | 25.8 | 66.5 | 55.5 | 52.5 |
| Ours | 3 | $\approx 5$ | 80.2 | 31.8 | 29.8 | 66.4 | 58.8 | 53.4 |
| GRPO (RL) | 2 | $\approx 190$ | 24.4 | 0.5 | 11.6 | 29.5 | 24.0 | 18.0 |
| Ours | 2 | $\approx 5$ | 45.0 | 2.7 | 10.6 | 34.9 | 29.2 | 24.5 |

Table 8: Comparison of first-step and fine-tuning (FT) cost along with downstream accuracy under 3-bit and 2-bit quantization on Qwen3-1.7B. We report training time normalized to a single H100 GPU.

| Method | Bit Width | First step | FT | FT Epochs | FT Sequences | Avg. Acc. |
|--------|-----------|------------|----|-----------|--------------|-----------|
| EfficientQAT | 3 | $\approx 4$ | $\approx 22$ | 1 | 32k | 53.5 |
| BitDistiller | 3 | $\approx 0.25$ | $\approx 26$ | 1 | 32k | 39.2 |
| Block+GRPO | 3 | $\approx 4$ | $\approx 1180$ | 1 | 32k | N/A |
| Ours | 3 | $\approx 4$ | $\approx 23$ | 1 | 32k | 55.2 |
| EfficientQAT | 2 | $\approx 4$ | $\approx 66$ | 3 | 32k | 18.3 |
| BitDistiller | 2 | $\approx 0.25$ | $\approx 78$ | 3 | 32k | 15.2 |
| Block+GRPO | 2 | $\approx 4$ | $\approx 1016$ | 3 | 32k | N/A |
| Ours | 2 | $\approx 4$ | $\approx 80$ | 3 | 32k | 28.4 |

## A    COMPARISON WITH ON-POLICY REINFORCEMENT LEARNING

While on-policy reinforcement learning incurs substantial training overhead due to its autoregressive text generation, it is commonly observed that such methods can generalize well on reasoning tasks by learning from the model's own output distribution. Our approach, instead, doesn't require autoregressive text generation because it only uses sequences sampled from datasets. Given these contrasting properties, it is important to assess whether our proposed off-policy objective can match the generalization typically associated with on-policy methods.

To clarify this trade-off, we conduct a direct comparison between our proposal and on-policy reinforcement learning. Specifically, we compare GRPO (Shao et al., 2024), an SOTA on-policy RL approach, with our off-policy loss. To ensure a fair comparison, we conduct the experiment using 6K sequences. Following DeepSeek-R1 (Guo et al., 2025), we adopt accuracy and formatting rewards for GRPO. We filtered the OpenThoughts-1.2M dataset to retain only samples with extractable answer labels to ensure that the accuracy reward over the training dataset is well-defined before randomly sampling 6K sequences.

As shown in Table 7, our method achieves comparable performance at 3-bit precision and outperforms GRPO at 2-bit precision with significantly reduced training time. These results demonstrate that our off-policy approach is both training-efficient and particularly effective in low-bit settings.

## B    TRAINING TIME ANALYSIS

As observed in Section 4, our approach performs well on reasoning benchmarks. Despite this promising performance, the overall training cost is also critical for practical quantization. This section compares the training time in both the first and fine-tuning stages to clarify this consideration.

Table 8 compares the required training time for Qwen3-1.7B with other QAT approaches. The results demonstrate that our method achieves a favorable trade-off between accuracy and training time, particularly under aggressive low-bit quantization. Notably, in the 2-bit setting, our approach achieves over 10% higher accuracy than EfficientQAT, while maintaining comparable training efficiency.

Table 9: Generation length comparison of Qwen3 models under different quantization bit-widths.

| Model | Bit Width | MATH-500 | | | GPQA-diamond | | | LiveCodeBench |
|---|---|---|---|---|---|---|---|---|
| | | Overall | Correct | Incorrect | Overall | Correct | Incorrect | Overall |
| Qwen3-1.7B | 16 | 5905 | 4706 | 15610 | 6744 | 5561 | 7480 | 12329 |
| Qwen3-1.7B | 3 | 7325 | 5059 | 17247 | 8564 | 7168 | 9201 | 15309 |
| Qwen3-1.7B | 2 | 13803 | 6483 | 21065 | 14497 | 11539 | 14944 | 22466 |
| Qwen3-8B | 16 | 5513 | 4923 | 14753 | 7397 | 6252 | 9235 | 11100 |
| Qwen3-8B | 3 | 6296 | 5359 | 16776 | 10473 | 8730 | 11896 | 14824 |
| Qwen3-8B | 2 | 9966 | 6136 | 25679 | 15738 | 13833 | 16779 | 20373 |

## C  ANALYSIS OF DECODING LENGTH

Analyzing additional reasoning metrics, such as token count, is important for understanding the qualitative behavior of quantized models. Differences in token usage induced by quantization may reveal shifts in reasoning style or depth that are not captured by standard benchmark accuracies. Evaluating these aspects provides a more comprehensive view of how quantization affects the reasoning process.

This section addresses this aspect by analyzing the decoding length across several reasoning benchmarks. As shown in Table 9, we find that the number of thinking tokens increases as the number of bits decreases. This increase in token length primarily arises from more frequent self-correction behavior. In other words, quantized models tend to revisit and validate their answers more frequently, as illustrated in Table 10. These results indicate that quantization can introduce redundant corrective reasoning. Thus, mitigating such unnecessary thought processes during QAT is important for controlling the additional thinking tokens induced by quantization.

## D  FUTURE WORK

This section discusses several promising directions for further enhancing the quantized reasoning models.

**Adaptive Mixing Ratio:**   Adapting the dataset mixing ratio in both the first and second stages is a promising research direction. This flexible strategy can enable more effective dataset selection tailored to each training phase, potentially improving overall model performance. In particular, dynamically adjusting the proportion of supervised, preference, or reasoning-heavy data may help stabilize training and reduce overfitting to any single data source. Developing principled methods for automatically determining these ratios could further enhance the efficiency and robustness of quantization-aware training. Additionally, adjusting the domain proportions within the reasoning dataset, such as increasing the ratio of code-related samples, may further enhance performance on more complex tasks like code generation. Exploring domain-specific mixing strategies remains an interesting avenue for future work.

**Increasing Model Size:**   Applying our approach on over 10B models is also a promising research direction for future research. While our current experiments focus on smaller-scale models, where the performance degradation of quantization is more severe, our promising results suggest that the benefits of our method may scale favorably with increased parameters. Due to the limited availability of computational resources, it is difficult to conduct such larger-scale experiments. Nevertheless, we believe that our approach will scale effectively to larger LLMs.

**Reducing Redundant Thinking:**   As observed in Appendix C, quantized models introduce additional reasoning tokens. From a practical perspective, this increase in tokens reduces throughput and requires larger KV-cache memory. Moreover, a quantitative analysis of generated samples reveals that quantization tends not to make models think more carefully, but instead leads them to repeatedly verify answers that were already produced. Suppressing such redundant verification behavior may enable more efficient quantized models.

Table 10: Reasoning text sample from 2bit Qwen3-8B quantized by our framework. The model can be seen revisiting its reasoning multiple times.

```
<think>
Okay, so I need to find the length of DE in this diagram. The problem
    says that sin D is 0.7. Let me try to visualize the diagram first.
    There's a right angle mark at D, E, and F, so maybe triangle D-E-F
    is a right triangle? Let me check the Asymptote code.

The Asymptote code defines points D, E, F. D is at (sqrt(51), 7), E
    is at (0, 7), and F is at (0, 0). Then they draw D-E-F and a right
    angle mark at D, E, F. Wait, but right angles are usually marked
    at a single vertex. Maybe it's a right triangle with right angle
    at D? Let me think. If D is connected to E and F, and there's a
    right angle mark at D, E, F, that might mean that the triangle is
    right-angled at D? But then E and F would be connected. Let me
    sketch mentally.

Point D is at (sqrt(51), 7), E is at (0, 7), so E is directly to the
    left of D on the horizontal line y=7. Then F is at (0, 0), which
    is directly below E on the vertical line x=0. So connecting D to E
    to F to D forms a triangle. Let me check coordinates:

- D: (sqrt(51), 7)
- E: (0, 7)
- F: (0, 0)

So DE is from D to E. Let me compute DE first. The coordinates of D
    are (sqrt(51), 7) and E is (0, 7). So the distance between D and E
    is sqrt[(sqrt(51) - 0)^2 + (7 - 7)^2] = sqrt[(sqrt(51))^2 + 0] =
    sqrt(51). Wait, sqrt(51) is approximately 7.07. But the problem
    says sin D is 0.7. Maybe I need to find DE using that? Wait, but
    maybe the diagram is not to scale? Or maybe there's some triangle
    where angle D is involved?

Wait, the Asymptote code draws D-E-F as a triangle and marks a right
    angle at D, E, and F? But (... 192 words omitted)

Wait, in triangle DEF, angle at D is between DE and DF. Let me
    confirm. If the right angle is (... 17 words omitted)

First, let me compute DE, EF, DF.

DE is from D to E: sqrt( (sqrt(51) - 0)^2 + (7 - 7)^2 ) = sqrt(51)
    7.07.

EF is from E to F: sqrt( (0 - 0)^2 + (7 - 0)^2 ) = 7.

DF is from D to F: sqrt( (sqrt(51) - 0)^2 + (7 - 0)^2 ) = sqrt(51 +
    49) = (... 3 words omitted)

So triangle DEF has sides DE = sqrt(51), EF = 7, DF = 10. Wait, but
    if the right angle is at E, then by Pythagoras, DE^2 + EF^2 should
    equal DF^2. Let's check:

DE^2 + EF^2 = 51 + 49 = 100, which is equal to DF^2 = 10^2 = 100.
    Perfect! So (... 33 words omitted)

Wait, the problem says sin D = 0.7. So sin(angle D) = 0.7. Let me
    verify if this is consistent (... 3 words omitted)
```

In triangle DEF, angle D is between DE and DF. So in that triangle,
    sin(angle D) = opposite side / hypotenuse. Wait, opposite side to
    angle D would be EF. Wait, in triangle DEF, angle at D: the side
    opposite angle D is EF, which is 7. The hypotenuse is DF = 10. So
    sin(angle D) = EF / DF = 7 / 10 = 0.7. Oh! That's exactly 0.7. So
    that matches the problem statement. Therefore, DE is sqrt(51), but
    the problem says sin D = 0.7, but DE is already given? Wait, but
    the problem is asking for DE? Wait, hold on, maybe I misread the
    problem?

Wait, the problem says "Suppose sin D = 0.7 in the diagram below.
    What is DE?" But in the Asymptote code, DE is sqrt(51). But maybe
    the diagram is not to scale? Wait, but the coordinates are given.
    Let me check again. Wait, maybe the diagram is different? Wait,
    the Asymptote code defines D as (sqrt(51), 7), E as (0, 7), F as
    (0, 0). So DE is sqrt( (sqrt(51) - 0)^2 + (7 - 7)^2 ) = sqrt(51).
    So DE is sqrt(51). But then why is sin D given as 0.7? Because in
    the triangle DEF, sin D is 7/10 = 0.7, so that's consistent.
    Therefore, is the answer DE = sqrt(51)? But maybe the problem is
    not referring to triangle DEF? Maybe there is another triangle in
    the diagram? Wait, the Asymptote code draws D-E-F as a triangle,
    but there's also a right angle mark at D, E, F? Wait, maybe the
    diagram is different?

But regardless, if the triangle DEF is right-angled at E, with DE =
    sqrt(51), EF = 7, DF = 10, then sin D is 7/10 = 0.7, which matches
    the problem statement. Therefore, DE is sqrt(51). But maybe the
    problem is not referring to triangle DEF? Wait, the problem says
    "in the diagram below", but the Asymptote code is given. Wait,
    maybe there's another triangle in the diagram? Let me check again.

But regardless, if the triangle DEF is right-angled at E, with DE =
    sqrt(51), EF = 7, DF = 10, (... 52 words omitted)

Wait, the Asymptote code draws D-E-F, which is triangle DEF, and then
    draws a right angle mark at D, E, (... 120 words omitted)

But regardless, the problem says sin D = 0.7, and in triangle DEF,
    sin D is 0.7. Therefore, DE is (... 219 words omitted)

Alternatively, maybe the diagram is not triangle DEF, but another
    triangle involving D and E? For example, maybe triangle
    D-E-something (... 83 words omitted)

Wait, but the problem says "Suppose sin D = 0.7 in the diagram below.
    What is DE?" If the diagram (... 41 words omitted)

```
pair D,E,F;
F = (0,0);
D = (sqrt(51),7);
E = (0,7);
draw(D--E--F--D);
draw(rightanglemark(D,E,F,15));
label("$D$",D,NE);
label("$E$",E,NW);
label("$F$",F,SW);
label("$7$",(E+F)/2,W);
```

So, in the diagram, there is a right angle at D, E, and F? Wait, no.
    The rightanglemark is drawn (... 159 words omitted)

```
Wait, but maybe the problem is not referring to triangle DEF, but
    another triangle? Let me think. The Asymptote code (... 101 words
    omitted)

But maybe the problem is not about triangle DEF? Wait, if angle D is
    given as sin D = 0.7, (... 96 words omitted)

Wait, let me check the problem again: "Suppose sin D = 0.7 in the
    diagram below. What is DE?" So (... 86 words omitted)

Wait, let me think again. If angle D is in triangle DEF, with sin D =
    0.7, and we can (... 171 words omitted)

Wait, maybe the diagram is not triangle DEF? Maybe there's a
    different triangle where angle D is involved. Wait, let (... 116
    words omitted)

But maybe the problem is not referring to triangle DEF? Maybe there's
    another triangle where DE is the hypotenuse or (... 38 words
    omitted)

Wait, but the Asymptote code draws only D-E-F. So unless there is
    another point, maybe the diagram is different? Wait, (... 137
    words omitted)

But in that case, we need to know EF, DF, and angle D. Wait, but if
    we have sin D (... 69 words omitted)

Wait, but if we don't have other sides, how can we find DE? Wait,
    unless there is a way to (... 181 words omitted)

But the problem says "in the diagram below", which is given by the
    Asymptote code. Therefore, the answer is sqrt(51). (... 153 words
    omitted)

Alternatively, maybe there is a misunderstanding in the problem.
    Maybe the diagram is not triangle DEF, but another figure? For
    (... 139 words omitted)

But maybe the problem is expecting a different approach? Let me think
    again. If sin D = 0.7, and we need to find DE. If we don't have
    coordinates, how would we do it? Let me consider triangle D-E-F,
    with right angle at E. Then, angle D is between DE and DF. So sin
    D = opposite / hypotenuse = EF / DF = 7 / 10 = 0.7. Therefore, if
    we know EF = 7, DF = 10, then DE can be found via Pythagoras: DE^2
    + 7^2 = 10^2 => DE^2 = 100 - 49 = 51 => DE = sqrt(51). Therefore,
    regardless of sin D, DE is sqrt(51). Therefore, the answer is
    sqrt(51). So maybe the problem is just giving sin D to confirm
    that the triangle is consistent, but the actual answer is
    sqrt(51). Therefore, the answer is \boxed{\sqrt{51}}.

But let me check once more. If the problem had not given coordinates,
    but only sin D = 0.7 and other information, how would we find DE?
    For example, if we had a right triangle with angle D, and we know
    sin D, and maybe another side, then we could find DE. But in this
    case, we have a right triangle with right angle at E, so we can
    use Pythagoras. So unless there is more information, the answer is
    sqrt(51). Therefore, I think that is the answer.

**Final Answer**
\boxed{\sqrt{51}}
</think>
```

