# OpenReview forum: "Towards Quantization-Aware Training for Ultra-Low-Bit Reasoning LLMs"
_ICLR.cc/2026/Conference — ICLR 2026 Poster_

### Official Review · Reviewer_d4eQ · 2025-10-27

**Soundness:** 2
**Presentation:** 3
**Contribution:** 2
**Rating:** 6
**Confidence:** 4

**Summary:**

This paper proposes a reasoning-oriented quantization-aware training (QAT) framework for ultra-low-bit (<4-bit) large language models (LLMs). The method introduces a two-stage pipeline combining mixed-domain calibration (pre-training + reasoning data) and a teacher-guided reward-rectification loss to preserve reasoning capability. Experiments on Qwen3 models (1.7B–8B) show substantial gains over PTQ baselines (GPTQ, AWQ) and competitive performance against BitNet-2B4T with far fewer training tokens.

**Strengths:**

Addresses an important and timely problem: maintaining reasoning ability under aggressive quantization.

Clear motivation and well-structured method.

Extensive empirical validation across reasoning benchmarks.

Ablation studies convincingly support the proposed components.

**Weaknesses:**

The approach is only evaluated up to 8B models; scaling to 70B+ remains unverified, though critical for real deployment.

The paper focuses on accuracy but lacks quantitative analysis of training cost and hardware efficiency (e.g., fine-tuning time, energy, or throughput gains).

Some comparisons (e.g., to full-training QAT methods like BitDistiller or Spectra) could be expanded to clarify relative trade-offs.

**Questions:**

How does the proposed teacher-guided loss scale computationally when the teacher is a large fp16 model?

Could mixed-domain ratios be dynamically adapted during training?

---

> ### Author Response · Authors · 2025-11-22
> **Response to Reviewer d4eQ**
>
> We thank the reviewer for the insightful comments and suggestions. We are encouraged that the reviewer found the motivation and empirical validation strong. The reviewer also raised several critical points regarding the need for additional comparison, throughput measurement, training time report, and further scaling. We appreciate this constructive feedback and have addressed each point. Below, we provide detailed responses to each concern.
>
> ---
> > **Some comparisons (e.g., to full-training QAT methods like BitDistiller or Spectra) could be expanded to clarify relative trade-offs.**
>
> Following the reviewer’s suggestion, we added comparison against two SOTA QAT approaches, EfficientQAT and BitDistiller, and found noticeable advantages across reasoning benchmarks. Notably, at 2-bit quantization, where existing methods struggle, our approach achieves 28.4% average accuracy compared to 18.3% and 15.2% for the baselines. At 3-bit, our method also outperforms both baselines (55.2% vs. 53.5% and 39.2%). We did not evaluate Spectra as it was not trained with reasoning-specific objectives, making direct comparison on reasoning benchmarks unfair to that method.
>
> Specifically, we apply SOTA QAT methods on Qwen3-1.7B using the same calibration data, the same fine-tuning dataset (32K sequences from OpenThoughts), and the same number of epochs to ensure a fair comparison. The table below provides the detailed results.
>
> | QAT | #bits | MATH-500 | LiveCodeBench | GPQA-Diamond | MMLU-Redux | IFEVAL | Avg. |
> | --- | --- | --- | --- | --- | --- | --- | --- |
> | EfficientQAT | 3 | 80.8 | 29.8 | 30.3 | 66.2 | 60.4 | 53.5 |
> | BitDistiller | 3 | 59.4 | 10.2 | 17.7 | 56.6 | 52.3 | 39.2 |
> | Ours | 3 | **82.7** | **33.0** | **31.7** | **67.7** | **61.0** | **55.2** |
> | EfficientQAT | 2 | 24.8 | 0.6 | 12.1 | 29.1 | 25.1 | 18.3 |
> | BitDistiller | 2 | 12.2 | 1.0 | **21.7** | 22.4 | 19.0 | 15.2 |
> | Ours | 2 | **48.6** | **6.5** | 14.5 | **40.1** | **32.2** | **28.4** |
>
> We further conducted additional experiments to compare the loss function of BitDistiller and our approach, as BitDistiller introduces confidence-aware KL divergence (CAKLD) loss, an extension of the KL divergence loss. To isolate the contribution of the loss function itself, we fine-tuned from identical block-wise-calibrated models. As shown in the table below, our loss formulation still outperforms in average accuracy on five benchmarks.
>
> | Loss | #bits | MATH-500 | LiveCodeBench | GPQA-Diamond | MMLU-Redux | IFEVAL | Avg. |
> | --- | --- | --- | --- | --- | --- | --- | --- |
> | CAKLD | 3 | 80.8 | 29.4 | 24.2 | 65.7 | 59.9 | 52.0 |
> | Ours loss | 3 | **82.7** | **33.0** | **31.7** | **67.7** | **61.0** | **55.2** |
> | CAKLD | 2 | 9.4 | 0.3 | **15.2** | 31.8 | 22.2 | 15.8 |
> | Ours loss | 2 | **48.6** | **6.5** | 14.5 | **40.1** | **32.2** | **28.4** |
>
> We believe these additional comparisons strengthen our paper by demonstrating that our proposed framework significantly outperforms existing SOTA QAT methods on reasoning benchmarks, a critical capability for modern LLMs. Particularly at 2-bit quantization, our approach achieves over 10% improvement in average accuracy. These results help validate both the necessity and effectiveness of our proposed approach for extremely low-bit quantization for reasoning models.
>
> ---
> > **The approach is only evaluated up to 8B models; scaling to 70B+ remains unverified, though critical for real deployment.**
>
> We acknowledge that evaluating larger models (70B+) would provide valuable insights into scalability for deployment scenarios. However, we respectfully note that our evaluation of models with up to 8B parameters still provides meaningful contributions, as increasing the number of parameters generally leads to greater robustness to quantization due to increased parameter redundancy, as observed in prior work [Liu et al., arxiv:2504.04823]. Therefore, our method, which demonstrates strong effectiveness on smaller models, where quantization performance degradation is severe, is likely to perform favorably on larger architectures as well. That said, we fully agree that verification on 70B+ models would strengthen the work. While resource limitations prevented us from evaluating 70B+ models in this submission, we acknowledge this as important future work and plan to report these results in an extended version.
>
> ---
> > **How does the proposed teacher-guided loss scale computationally when the teacher is a large fp16 model?**
>
> Fine-tuning for 2-bit Qwen3-8B takes around 266 H100 GPU hours, while 2-bit Qwen3-1.7B takes around 80 H100 GPU hours. The training time scales approximately 3.3× for a 4.7× increase in model size, demonstrating sub-linear scaling with respect to the target model size.

---

> ### Author Response · Authors · 2025-11-22
> **Response to Reviewer d4eQ**
>
> > **The paper focuses on accuracy but lacks quantitative analysis of hardware efficiency.**
>
> The primary focus of the paper is to realize reasoning LLMs with extremely low-bit representations, and our contribution is to demonstrate that high-quality 2- or 3-bit reasoning models are achievable, despite existing methods having failed to deliver.
>
> As our framework doesn’t rely on the specific low-bit numerical representations, the inference latency or memory footprint gains are inherited from prior quantization research. In this work, we employ a quantization function that is fully compatible with the GPTQ format.
>
> To verify this compatibility, we measured inference metrics on an A6000 GPU using BitBLAS framework. The results on Qwen3-8B are summarized in the table below.
>
> | Method | Config  | # Bit | Latency (s) | Throughput (tokens/s) | Peak Mem (GB) |
> | --- | --- | --- | --- | --- | --- |
> | GPTQ | 1024in_128out | 4 | 10.3 | 12.4 | 9.3 |
> | GPTQ | 1024in_128out | 2 | 9.9 | 12.9 | 6.0 |
> | Ours | 1024in_128out | 2 | 9.4 | 13.6 | 6.0 |
>
> These results demonstrate that our model achieves comparable inference characteristics. The slight latency difference stems from run-to-run timing variations.
>
> ---
> > **The paper focuses on accuracy but lacks quantitative analysis of training costs.**
>
> Following the reviewer's suggestion, we evaluated training efficiency. Our approach achieves superior accuracy while maintaining comparable training costs, especially in 2-bit quantization.
>
> The following table compares estimated training time across methods using identical training epochs and sequences. These results demonstrate that our framework achieves higher accuracy with comparable training resources, establishing a favorable  trade-off between accuracy and efficiency.
>
> Since the authors of EfficientQAT report that their method achieves approximately 2/3 of BitDistiller's training time, our current implementation of EfficientQAT may still have room for optimization. However, even accounting for this implementation difference, our method provides an attractive trade-off between accuracy and training cost, achieving substantially higher performance with reasonable computational overhead.
>
> |Method  | #bits | First step (H100 GPU hours) | Fine-tuning (H100 GPU hours)  | Fine-tuning Epochs | Fine-tuning Sequences | Avg. Acc. |
> | --- | --- | --- | --- | --- | --- | --- |
> | EfficientQAT | 3 | approx. 4 | 22 | 1 | 32k | 53.5 |
> | BitDistiller | 3 | approx. 0.25 | 26 | 1 | 32k | 39.2 |
> | Block+GRPO | 3 | approx. 4 | 1180 | 1 | 32k | N/A |
> | Ours | 3 | approx. 4 | 23 | 1 | 32k | **55.2** |
> | EfficientQAT | 2 | approx. 4 | 66 | 3 | 32k | 18.3 |
> | BitDistiller | 2 | approx. 0.25 | 78 | 3 | 32k | 15.2 |
> | Block+GRPO | 2 | approx. 4 | 1016 | 3 | 32k | N/A |
> | Ours | 2 | approx. 4 | 80 | 3 | 32k | **28.4** |
>
> > **Could mixed-domain ratios be dynamically adapted during training?**
>
> While we have not explored dynamic adaptation in our current work, we believe this idea is a promising direction. Dynamically adjusting the calibration ratio based on validation performance or quantization sensitivity could further improve the balance between reasoning ability and general knowledge. We will include this as a future research direction in our discussion.

---

### Official Review · Reviewer_Ncvd · 2025-10-31

**Soundness:** 3
**Presentation:** 3
**Contribution:** 3
**Rating:** 6
**Confidence:** 3

**Summary:**

This paper proposes a reasoning-oriented quantization-aware training (QAT) pipeline for ultra-low-bit (2–3 bit) LLMs.
The key idea is that reasoning abilities (acquired in post-training) and commonsense knowledge (from pre-training) respond differently to quantization.
To address this, the authors design a two-stage QAT framework:

Mixed-domain calibration (80% reasoning data + 20% pre-training data) for balanced preservation of knowledge and reasoning.

Teacher-guided reward-rectification fine-tuning, a reinforcement-inspired objective that restores reasoning accuracy efficiently.

**Strengths:**

The paper addresses an important and underexplored problem—maintaining reasoning capabilities during ultra-low-bit quantization.

The domain-sensitivity analysis (reasoning vs. pre-training) is interesting and helps motivate the mixed calibration design.

The pipeline is easy to follow and grounded in recent literature; figures and tables are clear and convincing.

**Weaknesses:**

The paper emphasizes accuracy but lacks analysis of training cost, throughput, and latency during inference—key aspects for practical quantization.

The 80/20 calibration ratio is fixed; it would be useful to see sensitivity analysis across ratios or datasets to validate robustness.

Other reasoning metrics, such as thinking token count, is not analyzed, which might reveal qualitative differences induced by quantization.

**Questions:**

When evaluating reasoning LLMs, task accuracy alone may not capture the full effect of quantization. Could you also report metrics related to thinking tokens—for example, the number of intermediate reasoning or "thought" tokens generated during inference?
In particular, does quantization (especially at 2 bits) affect how long the model needs to “think” or the length of its reasoning chain before reaching an answer?

---

> ### Author Response · Authors · 2025-11-22
> **Response to Reviewer Ncvd**
>
> We thank the reviewer for the insightful comments and suggestions. We are encouraged that the reviewer found the problem statement and the clarity of the paper strong. The reviewer also raised several critical points regarding the need for quantitative analysis of the number of thinking tokens, training and inference time reports, and detailed dataset analysis. We appreciate this constructive feedback and have addressed each point. Below, we provide detailed responses to each concern.
>
> ---
> > **Other reasoning metrics, such as thinking token count, is not analyzed, which might reveal qualitative differences induced by quantization.**
>
> Thank you for the insightful comment. Based on the suggestion, we provided an additional analysis of output token count for reasoning tasks. The table below summarizes the results.
>
> | Model | #Bits | MATH-500 | MATH-500 | MATH-500 | GPQA-diamond | GPQA-diamond | GPQA-diamond | LiveCodeBench |
> | --- | --- | --- | --- | --- | --- | --- | --- | --- |
> |  |  | Overall | Correct | Incorrect | Overall | Correct | Incorrect | Overall |
> | Qwen3-1.7B | 16 | 5905 | 4706 | 15610 | 6744 | 5561 | 7480 | 12329 |
> | Qwen3-1.7B | 3 | 7325 | 5059 | 17247 | 8564 | 7168 | 9201 | 15309 |
> | Qwen3-1.7B | 2 | 13803 | 6483 | 21065 | 14497 | 11539 | 14944 | 22466 |
> | Qwen3-8B | 16 | 5513 | 4923 | 14753 | 7397 | 6252 | 9235 | 11100 |
> | Qwen3-8B | 3 | 6296 | 5359  | 16776 | 10473 | 8730 | 11896 | 14824 |
> | Qwen3-8B | 2 | 9966 | 6136 | 25679 | 15738 | 13833 | 16779 | 20373 |
>
> We find that the number of thinking tokens increases as the number of bits decreases. Interestingly, through qualitative analysis of the outputs, we observe that this increase stems not primarily from longer reasoning chains, but rather from increased self-correction behavior. In other words, quantized models tend to revisit and validate their answers more frequently. However, these repeated validation processes often appear redundant and may not consistently improve accuracy.
>
> These findings suggest that constraining output token budgets could be a promising direction for future QAT research to avoid redundant validation processes and improve inference efficiency. In the revised version, we will include the table and full outputs for sample questions to provide more detailed qualitative analysis.
>
> ---
> > **The paper emphasizes accuracy but lacks analysis of throughput, and latency during inference—key aspects for practical quantization.**
>
> The primary focus of the paper is to realize reasoning LLMs with extremely low-bit representations, and our contribution is to demonstrate that high-quality 2- or 3-bit reasoning models are achievable, despite existing methods having failed to deliver.
>
> As our framework doesn’t rely on the specific low-bit numerical representations, the inference latency or memory footprint gains are inherited from prior quantization research. In this work, we employ a quantization function that is fully compatible with the GPTQ format.
>
> To verify this compatibility, we measured inference metrics on an A6000 GPU using BitBLAS framework. The results on Qwen3-8B are summarized in the table below.
>
> | Method | Config  | # Bit | Latency (s) | Throughput (tokens/s) | Peak Mem (GB) |
> | --- | --- | --- | --- | --- | --- |
> | GPTQ | 1024in_128out | 4 | 10.3 | 12.4 | 9.3 |
> | GPTQ | 1024in_128out | 2 | 9.9 | 12.9 | 6.0 |
> | Ours | 1024in_128out | 2 | 9.4 | 13.6 | 6.0 |
>
> These results demonstrate that our model achieves comparable inference characteristics. The slight latency difference stems from run-to-run timing variations.

---

> ### Author Response · Authors · 2025-11-22
> **Response to Reviewer Ncvd**
>
> > **The paper emphasizes accuracy but lacks analysis of training cost.**
>
> Following the reviewer's suggestion, we evaluated training efficiency. Our approach achieves superior accuracy while maintaining comparable training costs, especially in 2-bit quantization.
>
> The following table compares estimated training time across methods using identical training epochs and sequences. These results demonstrate that our framework achieves higher accuracy with comparable training resources, establishing a favorable  trade-off between accuracy and efficiency.
>
> Since the authors of EfficientQAT report that their method achieves approximately 2/3 of BitDistiller's training time, our current implementation of EfficientQAT may still have room for optimization. However, even accounting for this implementation difference, our method provides an attractive trade-off between accuracy and training cost, achieving substantially higher performance with reasonable computational overhead.
>
> | Method | #bits | First step (H100 GPU hours) | Fine-tuning (H100 GPU hours)  | Fine-tuning Epochs | Fine-tuning Sequences | Avg. Acc. |
> | --- | --- | --- | --- | --- | --- | --- |
> | EfficientQAT | 3 | approx. 4 | 22 | 1 | 32k | 53.5 |
> | BitDistiller | 3 | approx. 0.25 | 26 | 1 | 32k | 39.2 |
> | Block+GRPO | 3 | approx. 4 | 1180 | 1 | 32k | N/A |
> | Ours | 3 | approx. 4 | 23 | 1 | 32k | **55.2** |
> | EfficientQAT | 2 | approx. 4 | 66 | 3 | 32k | 18.3 |
> | BitDistiller | 2 | approx. 0.25 | 78 | 3 | 32k | 15.2 |
> | Block+GRPO | 2 | approx. 4 | 1016 | 3 | 32k | N/A |
> | Ours | 2 | approx. 4 | 80 | 3 | 32k | **28.4** |
>
> ---
> > **The 80/20 calibration ratio is fixed; it would be useful to see sensitivity analysis across ratios or datasets to validate robustness.**
>
> We agree that validating the robustness of the calibration ratio is important. We have conducted this sensitivity analysis, and the results are presented in Figure 2 (page 3) of our paper. The figure demonstrates distinct trends between reasoning and non-reasoning benchmarks across different mixing ratios. We will add a clearer reference to this figure in the revised manuscript to improve readability.

---

### Official Review · Reviewer_b4un · 2025-11-02

**Soundness:** 2
**Presentation:** 2
**Contribution:** 2
**Rating:** 4
**Confidence:** 3

**Summary:**

This paper analyzes the challenge of preserving reasoning capability under ultra low bit quantization and proposes a mixed calibration data strategy together with a two stage QAT pipeline. The method mixes reasoning and pre training data for calibration and then uses a teacher guided reward rectification loss in fine tuning to restore reasoning ability.

**Strengths:**

The paper studies an important issue of maintaining reasoning capabilities under aggressive quantization.

The analysis reveals that reasoning data has higher sensitivity to quantization than pre training data, and the mixed calibration design is logically motivated.

**Weaknesses:**

The novelty appears limited. The performance gain mainly comes from applying distillation on reasoning data, as shown in Table 4, where the KL or distillation term dominates the improvement and the proposed reward rectification formulation contributes marginal benefit. Distillation is a existing technique in low bit QAT.

The experimental comparison is not convincing. The main table (table 2) only compares with GPTQ and AWQ which are not QAT methods. Fairness requires comparison with recent QAT approaches, particularly distillation based ones such as EfficientQAT and BitDistiller.

Some claims lack evidence. The paper states that RL methods incur large autoregressive training overhead, but there is no quantitative analysis or comparison against RL based fine tuning for QAT. Similarly the equivalence to on policy learning is mentioned, but no evidence is provided to show practical benefit over standard supervised tuning.

Even with distillation, the model still shows a significant drop from BF16 performance on challenging tasks such as LiveCodeBench, suggesting that reasoning capability is not fully preserved and further improvements are required.

**Questions:**

Even with strong distillation, ultra low bit models still lag FP16 on tasks like LiveCodeBench. Are there other opportunities to further narrow this gap?

Can the authors compare against efficient RL post training methods, for example DPO variants, to validate the claim that RL cost is prohibitive and the proposed loss is a more efficient substitute?

How does the training cost of the proposed method compare with EfficientQAT and BitDistiller in terms of GPU hours?

---

> ### Author Response · Authors · 2025-11-22
> **Response to Reviewer b4un**
>
> We thank the reviewer for the insightful comments and suggestions. We are encouraged that the reviewer found the problem statement and analysis strong. The reviewer also raised several critical points regarding the need for detailed analysis, additional comparison, and training time reports. We appreciate this constructive feedback and have addressed each point. Below, we provide detailed responses to each concern.
>
> ---
> > **The performance gain mainly comes from applying distillation on reasoning data, as shown in Table 4, where the KL or distillation term dominates the improvement and the proposed reward rectification formulation contributes marginal benefit. Distillation is a existing technique in low bit QAT.**
>
> While distillation is indeed an existing technique, the novelty of our proposal lies in the combined loss design that overcomes the fundamental limitations of distillation-only approaches at extremely low-bit quantizations.
>
> To support our claim, we conduct a deeper investigation into the individual contributions of each component in the fine-tuning stage. Specifically, we extended the ablation study in Table 4 (page 8) to include 2-bit quantization. All ablation configurations use identical hyperparameters. These additional results reveal that at 2-bit, KL divergence alone fails catastrophically (2.2% accuracy on MATH-500), while adding reward rectification recovers performance to 48.6%, with solid improvements observed at 3-bit as described in the table below.
>
> | ($\alpha$, $\beta$)  | # bits | MATH-500 | LiveCodeBench | GPQA-Diamond | MMLU-Redux | IFEVAL | Avg. |
> | --- | --- | --- | --- | --- | --- | --- | --- |
> | (1.0, 0.0) | 3 | 78.2 | 28.5 | 21.7 | 66.1 | 60.6 | 51.0 |
> | (0.0, 1.0) | 3 | **82.8** | 32.4 | 30.3 | 67.0 | **63.0** | 55.1 |
> | (0.2, 1.0) | 3 | 82.7 | **33.0** | **31.7** | **67.7** | 62.1 | **55.4** |
> | (1.0, 0.0) | 2 | 22.8 | 1.5 | **15.7** | 32.7 | 25.0 | 19.5 |
> | (0.0, 1.0) | 2 | 2.2 | 0.0 | 2.0 | 6.6 | 8.0 | 3.8 |
> | (0.2, 1.0) | 2 | **48.6** | **6.5** | 14.5 | **40.1** | **32.2** | **28.4** |
>
> This performance degradation with the KL divergence loss likely arises from the significant quantization error at 2-bit, which creates a large distribution gap between the quantized and original models. This gap leads to noisy, unreliable gradients from the KL divergence loss. Introducing reward rectification loss helps stabilize the gradients by propagating supervised signals from the teacher's reliable probability estimates.
>
> These results demonstrate that our novelty lies not in distillation alone, but in designing a combined loss that requires both components for effective optimization under ultra-low-bit quantization. While distillation is an established technique, the additional results clearly demonstrate its critical limitations at 2 bits (achieving only 2.2%), thereby providing strong evidence for both the novelty and effectiveness of our proposed method.

---

> ### Author Response · Authors · 2025-11-22
> **Response to Reviewer b4un**
>
> > **The experimental comparison is not convincing. The main table (table 2) only compares with GPTQ and AWQ which are not QAT methods. Fairness requires comparison with recent QAT approaches, particularly distillation based ones such as EfficientQAT and BitDistiller.**
>
> Following the reviewer’s suggestion, we added a comparison against two SOTA QAT approaches, EfficientQAT and BitDistiller. We found noticeable advantages across reasoning benchmarks through the analysis in Qwen3-1.7B. Notably, at 2-bit quantization, where existing methods struggle, our approach achieves 28.4% average accuracy compared to 18.3% and 15.2% for the baselines. At 3-bit, our method also outperforms both baselines (55.2% vs. 53.5% and 39.2%).
>
> Specifically, we apply SOTA QAT methods on Qwen3-1.7B using the same calibration data, the same fine-tuning dataset (32K sequences from OpenThoughts), and the same number of epochs to ensure a fair comparison. The table below provides the detailed results.
>
> | QAT | #bits | MATH-500 | LiveCodeBench | GPQA-Diamond | MMLU-Redux | IFEVAL | Avg. |
> | --- | --- | --- | --- | --- | --- | --- | --- |
> | EfficientQAT | 3 | 80.8 | 29.8 | 30.3 | 66.2 | 60.4 | 53.5 |
> | BitDistiller | 3 | 59.4 | 10.2 | 17.7 | 56.6 | 52.3 | 39.2 |
> | Ours | 3 | **82.7** | **33.0** | **31.7** | **67.7** | **61.0** | **55.2** |
> | EfficientQAT | 2 | 24.8 | 0.6 | 12.1 | 29.1 | 25.1 | 18.3 |
> | BitDistiller | 2 | 12.2 | 1.0 | **21.7** | 22.4 | 19.0 | 15.2 |
> | Ours | 2 | **48.6** | **6.5** | 14.5 | **40.1** | **32.2** | **28.4** |
>
> We further conducted additional experiments to compare the loss function of BitDistiller and our approach, as BitDistiller introduces confidence-aware KL divergence (CAKLD) loss, an extension of the KL divergence loss. To isolate the contribution of the loss function itself, we fine-tuned from identical block-wise-calibrated models. As shown in the table below, our loss formulation still outperforms in average accuracy on five benchmarks.
>
> | Loss | #bits | MATH-500 | LiveCodeBench | GPQA-Diamond | MMLU-Redux | IFEVAL | Avg. |
> | --- | --- | --- | --- | --- | --- | --- | --- |
> | CAKLD | 3 | 80.8 | 29.4 | 24.2 | 65.7 | 59.9 | 52.0 |
> | Ours loss | 3 | **82.7** | **33.0** | **31.7** | **67.7** | **61.0** | **55.2** |
> | CAKLD | 2 | 9.4 | 0.3 | **15.2** | 31.8 | 22.2 | 15.8 |
> | Ours loss | 2 | **48.6** | **6.5** | 14.5 | **40.1** | **32.2** | **28.4** |
>
> We believe these additional comparisons strengthen our paper by demonstrating that our proposed framework significantly outperforms existing SOTA QAT methods on reasoning benchmarks, a critical capability for modern LLMs. Particularly at 2-bit quantization, our approach achieves over 10% improvement in average accuracy. These results help validate both the necessity and effectiveness of our proposed approach for extremely low-bit quantization for reasoning models.
>
> ---
> > **Even with strong distillation, ultra low bit models still lag FP16 on tasks like LiveCodeBench. Are there other opportunities to further narrow this gap?**
>
> While our model shows a drop from 16-bit performance on LiveCodeBench, it substantially outperforms other QAT baselines on this challenging benchmark. For example, at 3-bit precision, our method achieves 33.0% compared to 29.8% (EfficientQAT) and 10.2% (BitDistiller), demonstrating significant improvement over existing QAT approaches.
>
> We note that our training data for fine-tuning (OpenThoughts-1.2M) was originally designed for higher-precision models and is heavily weighted toward mathematical reasoning tasks. Quantized models may require more training data for such challenging tasks like coding due to quantization operation, balancing the training  data domains could further improve performance on underrepresented tasks. We will include this discussion in the future work section of the revised manuscript, as data requirements for quantized models represent an important research direction.

---

> ### Author Response · Authors · 2025-11-22
> **Response to Reviewer b4un**
>
> > **Some claims lack evidence. The paper states that RL methods incur large autoregressive training overhead, but there is no quantitative analysis or comparison against RL based fine tuning for QAT. Similarly the equivalence to on policy learning is mentioned, but no evidence is provided to show practical benefit over standard supervised tuning.**
>
> Following the reviewer’s valuable suggestion, we compared our method against GRPO [Shao et al., arXiv:2402.03300], an SOTA online RL approach, and supervised fine-tuning (SFT). The results on Qwen3-1.7B demonstrate that our off-policy method is particularly effective under severe quantization. At 2-bit, our method significantly outperforms both GRPO and SFT. At 3-bit, our method achieves comparable performance to GRPO while being more training efficient, and substantially outperforms SFT in reasoning performance.
>
> We chose GRPO over PPO or DPO because (1) GRPO represents the current SOTA in online RL-based LLM training, serving as a stronger baseline than PPO, and (2) unlike DPO (an offline method), GRPO directly addresses the reviewer’s concern about online vs. off-policy comparison.
>
> To ensure fair comparison, we conducted this experiment using 6K sequences for all methods (GRPO, SFT, and Ours). Following DeepSeek-R1, we adopt accuracy and formatting rewards for GRPO. To ensure that the accuracy reward is well-defined, we filtered the OpenThoughts dataset to retain only samples with extractable answer labels, then randomly sampled 6K sequences from this filtered set. While our full experiments use 32K sequences, GRPO's substantial computational cost (approximately 190-220 H100 GPU hours for 6K sequences vs. 20 hours for our method) makes scaling to 32K infeasible (over 1000 GPU hours).
>
> As shown in the table below, our method achieves comparable performance at 3-bit precision and outperforms GRPO at 2-bit precision with significantly reduced training time. In addition, our approach improves accuracy in both 2-bit and 3-bit compared with SFT.
>
> | 2nd Step | #bits | GPU Hours (H100) | MATH-500 | LiveCodeBench | GPQA-Diamond | MMLU-Redux | IFEVAL | Avg. |
> | --- | --- | --- | --- | --- | --- | --- | --- | --- |
> | SFT | 3 | ~4 | 79.2 | 26.3 | 27.3 | 64.0 | 50.6 | 49.5 |
> | GRPO (RL) | 3 | ∼220 | **83** | 31.5 | 25.8 | **66.5** | 55.5 | 52.5 |
> | **Ours** | 3 | ∼5 | 80.2 | **31.8** | **29.8** | 66.4 | **58.8** | **53.4** |
> | SFT | 2 | ∼4 | 30.8 | 0.9 | **11.6** | 28.5 | 21.6 | 18.7 |
> | GRPO (RL) | 2 | ∼190 | 24.4 | 0.5 | **11.6** | 29.5 | 24.0 | 18.0 |
> | **Ours** | 2 | ∼5 | **45** | **2.7** | 10.6 | **34.9** | **29.2** | **24.5** |
>
> We believe these results strongly validate our approach by demonstrating two key advantages: (1) it addresses a clear limitation of online RL under severe quantization, and (2) even when online RL performs well, our method provides superior training efficiency while maintaining comparable accuracy. This establishes our off-policy approach as a more practical and effective solution for ultra-low-bit quantization of reasoning models.
>
> > **How does the training cost of the proposed method compare with EfficientQAT and BitDistiller in terms of GPU hours?**
>
> Following the reviewer's suggestion, we evaluated training efficiency. Our approach achieves superior accuracy while maintaining comparable training costs, especially in 2-bit quantization.
>
> The following table compares estimated training time across methods using identical training epochs and sequences. These results demonstrate that our framework achieves higher accuracy with comparable training resources, establishing a favorable  trade-off between accuracy and efficiency.
>
> Since the authors of EfficientQAT report that their method achieves approximately 2/3 of BitDistiller's training time, our current implementation of EfficientQAT may still have room for optimization. However, even accounting for this implementation difference, our method provides an attractive trade-off between accuracy and training cost, achieving substantially higher performance with reasonable computational overhead.
>
> |Method  | #bits | First step (H100 GPU hours) | Fine-tuning (H100 GPU hours)  | Fine-tuning Epochs | Fine-tuning Sequences | Avg. Acc. |
> | --- | --- | --- | --- | --- | --- | --- |
> | EfficientQAT | 3 | approx. 4 | 22 | 1 | 32k | 53.5 |
> | BitDistiller | 3 | approx. 0.25 | 26 | 1 | 32k | 39.2 |
> | Block+GRPO | 3 | approx. 4 | 1180 | 1 | 32k | N/A |
> | Ours | 3 | approx. 4 | 23 | 1 | 32k | **55.2** |
> | EfficientQAT | 2 | approx. 4 | 66 | 3 | 32k | 18.3 |
> | BitDistiller | 2 | approx. 0.25 | 78 | 3 | 32k | 15.2 |
> | Block+GRPO | 2 | approx. 4 | 1016 | 3 | 32k | N/A |
> | Ours | 2 | approx. 4 | 80 | 3 | 32k | **28.4** |

---

### Official Review · Reviewer_tqL3 · 2025-11-03

**Soundness:** 3
**Presentation:** 3
**Contribution:** 3
**Rating:** 4
**Confidence:** 4

**Summary:**

This paper addresses the significant challenge of performance degradation in large language models (LLMs) on reasoning tasks when subjected to ultra-low-bit (<4 bits) quantization. The authors hypothesize that this degradation stems from the fact that quantization disproportionately affects the complex knowledge structures introduced during post-training (e.g., SFT, preference optimization) compared to the commonsense knowledge acquired during pre-training.
To counter this, the paper proposes a novel two-stage Quantization-Aware Training (QAT) pipeline:

- Stage 1 (Calibration): Performs an initial block-wise quantization using a mixed-domain calibration dataset (80% reasoning, 20% pre-training) to preserve both reasoning capabilities and general knowledge.

- Stage 2 (Fine-Tuning): Fine-tunes the quantized model using a novel objective function called the "teacher-guided reward-rectification loss". This loss reweights the standard supervised fine-tuning (SFT) loss using the probabilities from the full-precision teacher model, combined with a KL divergence term. This is proposed as an efficient, offline alternative to RL-based methods.

Experiments on Qwen3 models show that this pipeline dramatically outperforms standard post-training quantization (PTQ) baselines like GPTQ and AWQ, especially at 2-bit precision, and also shows competitive results against the specialized BitNet-2B4T model.

**Strengths:**

- Clear Problem Statement and Novel Insight: The paper targets a critical problem (quantization failure on reasoning) and provides a well-articulated insight: the differential impact of quantization on pre-training vs. post-training knowledge (validated in Fig 2a).

- Well-Motivated Methodological Components:
  - The mixed-domain calibration (Stage 1) is a simple, effective, and well-supported solution (Table 1) to the domain-sensitivity problem.
  - The teacher-guided reward-rectification loss (Stage 2) is a clever modification of SFT that correctly identifies the unreliability of the quantized student's probabilities and leverages the stable, full-precision teacher.

- Impressive and Significant Empirical Results: The performance gains, especially for 2-bit quantization, are dramatic (Table 2). The method successfully retains performance (e.g., 55.07% avg for 8B) where SOTA PTQ methods (GPTQ, AWQ) fail completely (~3-5% avg). This is a significant practical achievement.

- Strong Ablation Studies (Section 4.4): The ablations provide excellent support for the method's design. Table 3 compellingly demonstrates that both the calibration stage (C) and the proposed reward-rectification loss (R) are essential for success.

**Weaknesses:**

- Regarding Baseline Comparisons (QAT vs. PTQ): A primary concern is the comparison between the proposed QAT method and the PTQ (GPTQ, AWQ) baselines. While the results are impressive, it is generally expected that QAT methods (which involve fine-tuning) will outperform PTQ methods, especially at ultra-low bit-widths. The paper would be more convincing if it included a direct comparison against other state-of-the-art QAT methods (like EfficientQAT or UPQ, which are cited). The authors mention that other methods target different settings, but the paper would be significantly strengthened if the authors could reproduce at least one of these QAT baselines in their own evaluation framework. This would provide a more direct, like-for-like comparison.

- Regarding the Justification for the Proposed Loss: The motivation for the "teacher-guided reward-rectification loss" is its efficiency compared to online RL methods. This is a valid point. However, we note that online, on-policy RL methods (like PPO) derive their generalization benefit from learning on the model's own distribution. The proposed method is an offline, off-policy approach. It would be very helpful if the authors could provide more discussion or empirical evidence to support the claim that this offline approach achieves comparable generalization to the online methods it seeks to replace. This is a central and very interesting claim, and validating it (perhaps with an ablation against a DPO or PPO baseline) would make the contribution even stronger.

- Minor Clarification on Training Data: There appears to be a slight inconsistency in the reporting of training data, which could be easily clarified. The abstract mentions "40K training sequences", Section 4.1 lists "32,768 samples", and Table 5 notes "0.2M" and "0.8M" tokens. We would appreciate it if the authors could clarify the relationship between these numbers in the final version to avoid potential confusion.

**Questions:**

- Regarding the baseline comparisons (Weakness 1), the authors are strongly encouraged to include a direct comparison against at least one SOTA QAT method (e.g., EfficientQAT, UPQ). This would substantially bolster the paper's claims.

- To validate the claims about generalization (Weakness 2), it would be highly beneficial to add an ablation study that compares the proposed off-policy loss to a standard online RL baseline, such as PPO or DPO, on the same reasoning benchmarks.

- In the ablation for the loss function (Table 4), the KL divergence term seems to be heavily weighted $(\alpha=0.2, \beta=1.0)$. This suggests much of the performance gain might come from simple distribution alignment. We would be interested to see more sensitivity analysis on the alpha parameter.

- Given that "deployment" is a key motivator, the paper would be more complete if it included practical metrics like inference latency (tokens/sec) or memory footprint measurements, which are currently missing.

---

> ### Author Response · Authors · 2025-11-22
> **Response to Reviewer tqL3**
>
> We thank the reviewer for the insightful comments and suggestions. We are encouraged that the reviewer found our problem statement compelling and our empirical results strong. The reviewer also raised several critical points regarding the need for baseline comparisons, evaluations of inference metrics, and a detailed analysis of the second-stage loss. We appreciate this constructive feedback and have addressed each point. Below, we provide detailed responses to each concern.
>
> ---
> > **The authors are strongly encouraged to include a direct comparison against at least one SOTA QAT method. This would substantially bolster the paper's claims**
>
> Following the reviewer’s suggestion, we added a comparison against two SOTA QAT approaches, EfficientQAT and BitDistiller. We found noticeable advantages across reasoning benchmarks through the analysis of Qwen3-1.7B. Notably, at 2-bit quantization, where existing methods struggle, our approach achieves 28.4% average accuracy compared to 18.3% and 15.2% for the baselines. At 3-bit, our method also outperforms both baselines (55.2% vs. 53.5% and 39.2%).
>
> Specifically, we apply SOTA QAT methods on Qwen3-1.7B using the same calibration data, the same fine-tuning dataset (32K sequences from OpenThoughts), and the same number of epochs to ensure a fair comparison. The table below provides the detailed results.
>
> | QAT | #bits | MATH-500 | LiveCodeBench | GPQA-Diamond | MMLU-Redux | IFEVAL | Avg. |
> | --- | --- | --- | --- | --- | --- | --- | --- |
> | EfficientQAT | 3 | 80.8 | 29.8 | 30.3 | 66.2 | 60.4 | 53.5 |
> | BitDistiller | 3 | 59.4 | 10.2 | 17.7 | 56.6 | 52.3 | 39.2 |
> | Ours | 3 | **82.7** | **33.0** | **31.7** | **67.7** | **61.0** | **55.2** |
> | EfficientQAT | 2 | 24.8 | 0.6 | 12.1 | 29.1 | 25.1 | 18.3 |
> | BitDistiller | 2 | 12.2 | 1.0 | **21.7** | 22.4 | 19.0 | 15.2 |
> | Ours | 2 | **48.6** | **6.5** | 14.5 | **40.1** | **32.2** | **28.4** |
>
> We further conducted additional experiments to compare the loss function of BitDistiller and our approach, as BitDistiller introduces confidence-aware KL divergence (CAKLD) loss, an extension of the KL divergence loss. To isolate the contribution of the loss function itself, we fine-tuned from identical block-wise-calibrated models. As shown in the table below, our loss formulation still outperforms in average accuracy on five benchmarks.
>
> | Loss | #bits | MATH-500 | LiveCodeBench | GPQA-Diamond | MMLU-Redux | IFEVAL | Avg. |
> | --- | --- | --- | --- | --- | --- | --- | --- |
> | CAKLD | 3 | 80.8 | 29.4 | 24.2 | 65.7 | 59.9 | 52.0 |
> | Ours loss | 3 | **82.7** | **33.0** | **31.7** | **67.7** | **61.0** | **55.2** |
> | CAKLD | 2 | 9.4 | 0.3 | **15.2** | 31.8 | 22.2 | 15.8 |
> | Ours loss | 2 | **48.6** | **6.5** | 14.5 | **40.1** | **32.2** | **28.4** |
>
> We believe these additional comparisons strengthen our paper by demonstrating that our proposed framework significantly outperforms existing SOTA QAT methods on reasoning benchmarks, a critical capability for modern LLMs. Particularly at 2-bit quantization, our approach achieves over 10% improvement in average accuracy. These results help validate both the necessity and effectiveness of our proposed approach for extremely low-bit quantization for reasoning models.

---

> ### Author Response · Authors · 2025-11-22
> **Response to Reviewer tqL3**
>
> >  **It would be highly beneficial to add an ablation study that compares the proposed off-policy loss to a standard online RL baseline**
>
> Following the reviewer’s valuable suggestion, we compared our method against GRPO [Shao et al., arXiv:2402.03300], an SOTA online RL approach. The results on Qwen3-1.7B demonstrate that our off-policy method is particularly effective under severe quantization: it significantly outperforms online RL at 2-bit quantization (24.5% vs 18.0% average accuracy) and achieves comparable performance with higher training efficiency at 3-bit quantization.
>
> We chose GRPO over PPO or DPO because (1) GRPO represents the current SOTA in online RL-based LLM training, serving as a stronger baseline than PPO, and (2) unlike DPO (an offline method), GRPO directly addresses the reviewer’s concern about online vs. off-policy comparison.
>
> To ensure fair comparison, we conducted this experiment using 6K sequences for all methods (GRPO and Ours). Following DeepSeek-R1, we adopt accuracy and formatting rewards for GRPO. To ensure that the accuracy reward is well-defined, we filtered the OpenThoughts dataset to retain only samples with extractable answer labels, then randomly sampled 6K sequences from this filtered set. While our full experiments use 32K sequences, GRPO's computational cost (approximately 190-220 H100 GPU hours for 6K sequences vs. 20 hours for our method) makes scaling to 32K infeasible (over 1000 GPU hours).
>
> As shown in the table below, our method achieves comparable performance at 3-bit precision and outperforms GRPO at 2-bit precision with significantly reduced training time.
>
> | 2nd Step | #bits | GPU Hours (H100) | MATH-500 | LiveCodeBench | GPQA-Diamond | MMLU-Redux | IFEVAL | Avg. |
> | --- | --- | --- | --- | --- | --- | --- | --- | --- |
> | GRPO (RL) | 3 | ∼220 | **83** | 31.5 | 25.8 | **66.5** | 55.5 | 52.5 |
> | **Ours** | 3 | ∼5 | 80.2 | **31.8** | **29.8** | 66.4 | **58.8** | **53.4** |
> | GRPO (RL) | 2 | ∼190 | 24.4 | 0.5 | **11.6** | 29.5 | 24.0 | 18.0 |
> | **Ours** | 2 | ∼5 | **45** | **2.7** | 10.6 | **34.9** | **29.2** | **24.5** |
>
> We believe these results strongly validate our approach by demonstrating two key advantages: (1) it addresses a clear limitation of online RL under severe quantization, and (2) even when online RL performs well, our method provides superior training efficiency while maintaining comparable accuracy. This establishes our off-policy approach as a more practical and effective solution for extremely low-bit quantization of reasoning models.
>
> ---
> > **The KL divergence term seems to be heavily weighted (a=0.2, b=1.0). This suggests much of the performance gain might come from simple distribution alignment**
>
> We would like to clarify that the small weight on reward rectification (α=0.2) does not reflect a marginal contribution. Rather, the reward rectification loss is essential to overcome the fundamental limitations of distillation-only approaches at extremely low bit parameters.
>
> To demonstrate this, we extended the ablation study in Table 4 (page 8) to include 2-bit quantization. All ablation configurations use identical hyperparameters. These additional results reveal that at 2-bit, KL divergence alone fails catastrophically (2.2% accuracy on MATH-500), while adding reward rectification recovers performance to 48.6%, with solid improvements observed at 3-bit as described in the table below.
>
> | ($\alpha$, $\beta$)  | # bits | MATH-500 | LiveCodeBench | GPQA-Diamond | MMLU-Redux | IFEVAL | Avg. |
> | --- | --- | --- | --- | --- | --- | --- | --- |
> | (1.0, 0.0) | 3 | 78.2 | 28.5 | 21.7 | 66.1 | 60.6 | 51.0 |
> | (0.0, 1.0) | 3 | **82.8** | 32.4 | 30.3 | 67.0 | **63.0** | 55.1 |
> | (0.2, 1.0) | 3 | 82.7 | **33.0** | **31.7** | **67.7** | 62.1 | **55.4** |
> |  |  |  |  |  |  |  |  |
> | (1.0, 0.0) | 2 | 22.8 | 1.5 | **15.7** | 32.7 | 25.0 | 19.5 |
> | (0.0, 1.0) | 2 | 2.2 | 0.0 | 2.0 | 6.6 | 8.0 | 3.8 |
> | (0.2, 1.0) | 2 | **48.6** | **6.5** | 14.5 | **40.1** | **32.2** |** 28.4** |
>
> This performance degradation with the KL divergence loss likely arises from the significant quantization error at 2-bit, which creates a large distribution gap between the quantized and original models. This gap leads to noisy, unreliable gradients from the KL divergence loss. Introducing reward rectification loss helps stabilize the gradients by propagating supervised signals from the teacher's reliable probability estimates.
>
> These results clearly demonstrate that reward rectification plays an essential role, particularly at 2-bit where it enables significant performance recovery. This improvement in accuracy cannot be attributed to simple distribution alignment.

---

> ### Author Response · Authors · 2025-11-22
> **Response to Reviewer tqL3**
>
> > **Given that "deployment" is a key motivator, the paper would be more complete if it included practical metrics like inference latency (tokens/sec) or memory footprint measurements, which are currently missing**
>
> The primary focus of the paper is to realize reasoning LLMs with extremely low-bit representations, and our contribution is to demonstrate that high-quality 2- or 3-bit reasoning models are achievable, despite existing methods having failed to deliver.
>
> As our framework doesn’t rely on the specific low-bit numerical representations, the inference latency or memory footprint gains are inherited from prior quantization research. In this work, we employ a quantization function that is fully compatible with the GPTQ format.
>
> To verify this compatibility, we measured inference metrics on an A6000 GPU using BitBLAS framework. The results on Qwen3-8B are summarized in the table below.
>
> | Method | Config  | # Bit | Latency (s) | Throughput (tokens/s) | Peak Mem (GB) |
> | --- | --- | --- | --- | --- | --- |
> | GPTQ | 1024in_128out | 4 | 10.3 | 12.4 | 9.3 |
> | GPTQ | 1024in_128out | 2 | 9.9 | 12.9 | 6.0 |
> | Ours | 1024in_128out | 2 | 9.4 | 13.6 | 6.0 |
>
> These results demonstrate that our model achieves comparable inference characteristics. The slight latency difference stems from run-to-run timing variations.
>
> ---
> > **Minor Clarification on Training Data**
>
> We would like to clarify the training data. We acknowledge that inconsistent use of “sequences”, “samples”, and “tokens” may have caused confusion. We will standardize our terminology by reporting training data primarily in terms of tokens.
>
> In addition, we found the calculation error in token counts in Table 5. We have recalculated all values correctly, and will update the table and abstract in the revision as follows:
>
> - training tokens for Qwen3 8B and 4B: 32k$\times$10k (fine-tuning tokens)+4k$\times$2k (calibration tokens) = 328M tokens
> - training tokens for Qwen3 1.7B: 32k$\times$10k$\times$3 (fine-tuning tokens)+4k$\times$2k (calibration tokens) = 968M tokens
> - In the abstract: using 968M tokens
>
> Different training tokens in Qwen 1.7B and other models stem from different fine-tuning epochs. We will add the more detailed experimental settings in the revised version.

---

### Author Response · Authors · 2025-11-27
**Update manuscript**

We sincerely thank all reviewers for their valuable feedback and constructive comments. We have provided individual responses to each reviewer addressing their specific concerns. Based on the feedback received, we have revised the manuscript accordingly.

Below, we summarize the main changes made to the manuscript.
Summary of Changes:
1. We have added the new Section 4.4 to provide a comparison with QAT approaches (related to questions from Reviewers tqL3, b4un, and d4eQ)
2. We have added int2 results to Table 6 to provide an in-depth ablation for the second-stage loss function (related to questions from Reviewers tqL3 and b4un)
3. We have added the new Appendix A to compare our method with on-policy reinforcement learning (related to questions from Reviewers tqL3 and b4un)
4. We have added the new Appendix B to provide a training time analysis (related to questions from Reviewers b4un, Ncvd, and d4eQ)
5. We have added the new Appendix C to analyze decoding length (related to questions from Reviewers Ncvd)
6. We have added the new Appendix D to discuss future work (related to questions from Reviewers b4un and d4eQ)
7. We have changed the number of training tokens in Table 5 and the abstract (related to questions from Reviewers tqL3)

We hope these revisions address the concerns raised. The additional results obtained during the rebuttal period—items (1) through items (4)—not only address the reviewer's concerns but also further strengthen the novelty of the paper and the effectiveness of the proposed method. Moreover, from the analysis of (5), we have derived new insights that may guide future research on quantization. Regarding the concern about throughput comparisons, we believe that ensuring compatibility with existing quantization formats is of primary importance and that this focus does not diminish the contributions of the proposed QAT algorithm.

We sincerely appreciate the reviewers' thoughtful feedback and the opportunity to strengthen our work.

---

### Author Response · Authors · 2025-12-03
**Final Remark to Area Chair**

Dear Area Chair,

We would like to express our gratitude and respect to the reviewers with the following remarks.
Unfortunately, due to the sudden change in the review process, there was insufficient time for further discussion with the reviewers. Nevertheless, we are grateful for the reviewers' dedicated feedback, which has not only helped us address the concerns raised but also strengthened our paper.

### **Summary of Rebuttal:**

**tqL3**

While the reviewer’s initial concerns were (1) lack of comparison with QAT approaches, (2) lack of comparison with online RL approaches, (3) small contribution to reasoning performance of teacher-guided reward rectification loss, and (4) lack of throughput evaluation, we believe these concerns have been adequately addressed through our additional experiments and clarifications in the rebuttal:

- (1) Comparison with EfficientQAT and BitDistiller, demonstrating superior performance over QAT approaches.
- (2) Comparison with GRPO, demonstrating superior performance with significantly less training time.
- (3) 2-bit ablation study, clarifying the significant contribution of the proposed loss.
- (4) Throughput comparison with GPTQ, showing compatibility with existing quantization optimizations.

**b4un**

While the reviewer’s initial concerns were (1) marginal contribution of teacher-guided reward rectification loss, (2) lack of comparison with QAT approaches, (3) lack of comparison with RL and SFT approaches, (4) remaining performance gap with FP16 on challenging tasks, we believe these concerns have been adequately addressed through our additional experiments and clarifications in the rebuttal:

- (1) 2-bit ablation study, clarifying the significant contribution of the proposed loss.
- (2)(3) Comparison with QAT, RL, and SFT baselines, confirming our method’s advantage.
- (4) Clarification on LiveCodeBench Performance, suggesting potential directions for improving performance on challenging tasks.

**Ncvd**

While the reviewer’s initial concerns were (1) lack of analysis of training and inference costs, (2) lack of sensitivity analysis across ratios or datasets, (3) lack of analysis of thinking token count, we believe these concerns have been adequately addressed through our additional experiments  and clarifications in the rebuttal:

- (1) Training and inference time analysis, confirming the practical efficiency of our method.
- (2) Remark on sensitivity analysis (Figure 2), confirming that single-domain calibration leads to degraded performance.
- (3) Analysis and insights on thinking token counts, revealing the increase in thinking tokens caused by quantization and its underlying factors.

**d4eQ**

While the reviewer’s initial concerns were (1) limited parameter scale, (2) lack of analysis of training and inference cost, (3) lack of comparison with QAT approaches, we believe these concerns have been adequately addressed through our additional experiments  and clarifications in the rebuttal:

- (1) Clarification on limited parameter scalability, demonstrating effectiveness under a more challenging setting with smaller models.
- (2) Training and inference time analysis, confirming the practical efficiency of our method.
- (3) Comparison with QAT baselines, demonstrating superior performance over QAT approaches.

In summary, all reviewers appreciated our well-motivated methodology and convincing empirical results. We believe our rebuttal has addressed the remaining concerns and further improved the paper.

---

### Meta-Review · Area_Chair_5NuM · 2026-01-06

**Summary:**

This paper proposes a reasoning-oriented quantization-aware training (QAT) framework for ultra-low-bit (2–3 bit) large language models, motivated by the observation that reasoning capabilities acquired during post-training are more sensitive to quantization than pre-training knowledge. The method introduces a two-stage pipeline combining mixed-domain calibration and a teacher-guided reward-rectification loss.

Across reviewers, the paper was consistently recognized as addressing an important and timely problem with clear motivation, a well-structured method, and strong empirical results, particularly at 2-bit quantization where existing methods largely fail. The main concerns raised in the reviews centered on (i) fairness and completeness of baseline comparisons (especially against QAT and RL-based methods), (ii) clarity of the contribution of the proposed loss beyond standard distillation, (iii) lack of training/inference efficiency analysis, (iv) missing comparisons to online RL and SFT approaches, (v) limited analysis of decoding behavior and calibration sensitivity, and (vi) scalability and deployment-related questions.

The rebuttal substantially strengthened the paper by adding direct comparisons with SOTA QAT methods (EfficientQAT, BitDistiller), comparisons with online RL (GRPO) and SFT, extended ablations at 2-bit precision, training cost and inference throughput analyses, thinking-token analysis, and calibration-ratio sensitivity results. Taken together, the revisions address the core technical concerns raised by reviewers and clarify both the novelty and practical relevance of the proposed approach, supporting an accept recommendation.

**Reviewer Concerns:**

### Concerns addressed by the rebuttal:

* **Lack of comparison with QAT baselines (tqL3, b4un, d4eQ):**
  The authors added direct, controlled comparisons with EfficientQAT and BitDistiller under identical calibration data, training data, and epochs. Results show consistent improvements, especially at 2-bit quantization, where the proposed method substantially outperforms prior QAT approaches.

* **Insufficient justification of the teacher-guided reward-rectification loss (tqL3, b4un):**
  Extended ablations, particularly at 2-bit, demonstrate that KL/distillation alone fails catastrophically, while adding reward rectification recovers large performance gains, clarifying that the contribution is not marginal and cannot be reduced to standard distillation.

* **Lack of comparison with online RL and SFT (tqL3, b4un):**
  The rebuttal includes comparisons with GRPO (online RL) and SFT, showing that the proposed off-policy approach achieves comparable or better accuracy with orders-of-magnitude lower training cost, especially under severe quantization.

* **Missing training and inference efficiency analysis (tqL3, Ncvd, d4eQ):**
  The authors added detailed training cost tables (GPU hours) and inference latency/throughput/memory measurements, demonstrating comparable efficiency to GPTQ while maintaining strong accuracy.

* **Calibration ratio sensitivity and robustness (Ncvd):**
  The paper already included ratio sensitivity analysis (Figure 2), and the rebuttal clarifies its interpretation, confirming that single-domain calibration degrades performance and supporting the mixed-domain design.

* **Lack of reasoning-behavior analysis (thinking tokens) (Ncvd):**
  The authors added a detailed analysis showing that lower-bit models produce more tokens due to increased self-correction, providing new insight into qualitative effects of quantization.

* **Baseline fairness and scope clarification (d4eQ):**
  Additional explanations justify why certain methods (e.g., Spectra) are not directly comparable, and expanded QAT comparisons clarify trade-offs.

* **Reporting inconsistencies and missing details (tqL3):**
  Training token counts and terminology were corrected and standardized in the revised manuscript.

* **Scaling beyond 8B models (d4eQ):**
  Evaluation is limited to models up to 8B parameters. While the authors argue smaller models are a harder quantization regime and cite prior work suggesting larger models are more robust, validation at 70B+ remains future work.

### Concerns partially resolved:

* **Dynamic calibration ratio adaptation (Ncvd, d4eQ):**
  The calibration ratio is fixed; adaptive strategies are acknowledged as a promising but unexplored direction.

**Reviewer Scores:**

* **Reviewer tqL3 (initial 4):** Likely remain unchanged or increase to **6**. The rebuttal directly addresses all major concerns, including QAT comparisons, RL baselines, loss ablations at 2-bit, efficiency analysis, and data reporting clarity.

* **Reviewer b4un (initial 4):** Likely remain unchanged or increase to **6**. The added QAT and RL comparisons, extended ablations demonstrating the necessity of reward rectification, and explicit training-cost analysis resolve the reviewer’s central objections regarding novelty and fairness.

* **Reviewer Ncvd (initial 6):** Likely unchanged. Training/inference cost analysis, thinking-token analysis, and calibration sensitivity directly address the reviewer’s questions.

* **Reviewer d4eQ (initial 6):** Likely unchanged. Added QAT comparisons, efficiency metrics, and scaling discussion address most concerns, with large-model scaling appropriately framed as future work.

Overall, the discussion and rebuttal materially strengthen the paper, resolve the key technical critiques, and support an **accept** recommendation.

---

### Decision · Program_Chairs · 2026-01-26

Accept (Poster)